# Light-driven dandelion-inspired microfliers

Yuanhao Chen[1], Cristian Valenzuela[1], Xuan Zhang[1], Xiao Yang[1], Ling Wang [1,2] ✉ & Wei Feng [1,2] ✉

In nature, many plants have evolved diverse flight mechanisms to disperse seeds by wind and propagate their genetic information. Inspired by the flight mechanism of the dandelion seeds, we demonstrate light-driven dandelion-inspired microfliers based on ultralight and super-sensitive tubular-shaped bimorph soft actuator. Like dandelion seeds in nature, the falling velocity of the as-proposed microflier in air can be facilely controlled by tailoring the degree of deformation of the "pappus" under different light irradiations. Importantly, the resulting microflier is able to achieve a mid-air flight above a light source with a sustained flight time of ~8.9 s and a maximum flight height of ~350 mm thanks to the unique dandelion-like 3D structures. Unexpectedly, the resulting microflier is found to exhibit light-driven upward flight accompanied by autorotating motion, and the rotation mode can be customized in either a clockwise or counterclockwise direction by engineering the shape programmability of bimorph soft actuator films. The research disclosed herein can offer new insights into the development of untethered and energy-efficient artificial aerial vehicles that are of paramount significance for many applications from environmental monitoring and wireless communication to future solar sail and robotic spacecraft.

Over the past two decades, autonomous unmanned aerial vehicles, and in particular fixed- and rotary-wing micro aerial vehicles (MAVs), have attracted significant interest due to their emerging applications in diverse civilian and military fields, such as covert surveillance, search and rescue, hazardous inspection, environmental monitoring, and aerial photography[1]. Conventional MAVs are composed of complex airframes with microcontrollers, rotary motors, batteries, and sensors, where the propulsion and actuation efficiency would become a challenging issue as the MAVs are further scaled down, due to the reduced power density of electromagnetic motors, decreased transmission efficiency, and increased influence of viscous effects as a result of reduced Reynolds number[2]. In this context, researchers have devoted extensive efforts to drawing inspiration from unprecedented insect aerodynamics for the development of insect-scale flapping-wing MAVs that can flap their tiny wings and create sufficient force to offset gravity for flight[3]. For example, insect-scale MAVs with flapping-wings capable of successfully performing tethered-controlled flight were demonstrated by using high-power-density piezoelectric artificial muscles and dielectric elastomer actuators[4,5]. Inspired by rhinoceros beetle,

untethered MAVs with deployable hindwings have been reported with the integration of miniaturized onboard electronics and power source[6]. It should be noted that the lack of lightweight and high-power-density batteries has greatly limited the untethered flight durations of insect-scale MAVs. Recently, wireless radio frequency power supply has been used to drive insect-scale flapping-wing MAVs[7], and light-powered untethered insect-sized flapping-wing MAVs have been developed by taking advantage of emerging photovoltaics and integrated electronics[8,9]. However, integrating insect-scale MAVs with complicated electronics or tethering them to an external power supply could lead to some unavoidable limitations, such as disturbed flight balance and extra weight, thus decreasing the flight efficiency. Therefore, it is highly desirable to explore a new strategy for the design and fabrication of insect-scale untethered MAVs with energetically efficient heavier-than-air flight.

In nature, wind-dispersed plant seeds, also known as diaspores, are known to provide an aerodynamic solution for developing untethered MAVs with lower energy consumption, efficient and passive flight, as well as maximum aerodynamic loading and minimal material

[1]School of Materials Science and Engineering, Tianjin University, Tianjin 300350, China. [2]Tianjin Key Laboratory of Composite and Functional Materials, Tianjin 300350, China. ✉e-mail: lwang17@tju.edu.cn; weifeng@tju.edu.cn

requirements[10–12]. As a representative example, the dandelion seeds (Taraxacum officinale agg.), composed of an achene and a bundle of around 100 bristly filaments called pappus, can travel as far as one kilometer in dry, windy, and warm conditions[13–15]. The pappus is spread out like an umbrella and functions as a drag-enhancing parachute during seed's flight, and it has been experimentally proved that the high porosity and precise spacing of the pappus promote the air flow passing through it, causing air to recycle in the pappus itself and creating a separate vortex ring[16], which is essential to provide convective updrafts for the flight of dandelion seeds and ensure that they remain aloft and disperse over a long distance. The unprecedented drag-enhancing configuration of natural dandelion seeds has been considered to be of particular advantage when implemented in power-free and wireless miniaturized MAVs. Recently, dandelion-inspired battery-free wireless sensing devices have been demonstrated to enable backscatter communication, wide-area dispersal (spanning 50–100 m), and upright landing with aerodynamic stability[17]. However, the resulting artificial dandelion devices show some limitations, such as high flight randomness and inadequate controllability. The development of dandelion-inspired artificial microfliers with robust controllability could shine new light on energy-efficient and untethered advanced microfliers that can be of paramount significance for many emerging applications such as environmental monitoring and wireless communication.

In this work, we report the design and fabrication of an untethered dandelion-inspired microflier (Fig. 1), which is spatially and temporally controlled by an ultralight and super-sensitive light-driven bimorph soft actuator composed of polyimide (PI) and low-density polyethylene (LDPE) layers that are electrostatically laminated and loaded with surface-functionalized ultrasmall gold nanorods (AuNRs). Compared to other reported soft actuators, the as-proposed bimorph soft actuator outperforms in several aspects, including excellent deformation derived from its large mismatch in coefficients of thermal expansion (CTEs), facile shape programmability, and outstanding sensitivity (e.g., hand temperature can cause large deformation)[18,19]. Taking aerodynamic lessons of drag-enhancing parachute flight in nature, the dandelion-inspired artificial microflier is then constructed from a tubular-shaped bimorph soft actuator film, of which one end is cut into thin strips and attached with ~40 strands of fiberglass ("pappus") (Fig. 1b–d). Thanks to the robust interface between PI and LDPE layers, large mismatch in their CTEs, and high/selective photothermal conversion of ultrasmall AuNRs, the "pappus" of dandelion-inspired microflier can be reversibly actuated from a closed state to an open state upon external light irradiation (Fig. 1e). Like dandelion seeds in nature, the falling velocity of artificial microfliers in air can be facilely controlled by tailoring the degree of deformation of the "pappus" under light irradiation with different intensities. Importantly, the unique tubular shape and dandelion-like structures could facilitate the formation of a separated vortex ring that underlies the flight of dandelion seeds, so that the microflier is able to achieve a mid-air flight above a light source with a sustained flight time of ~8.9 s and a

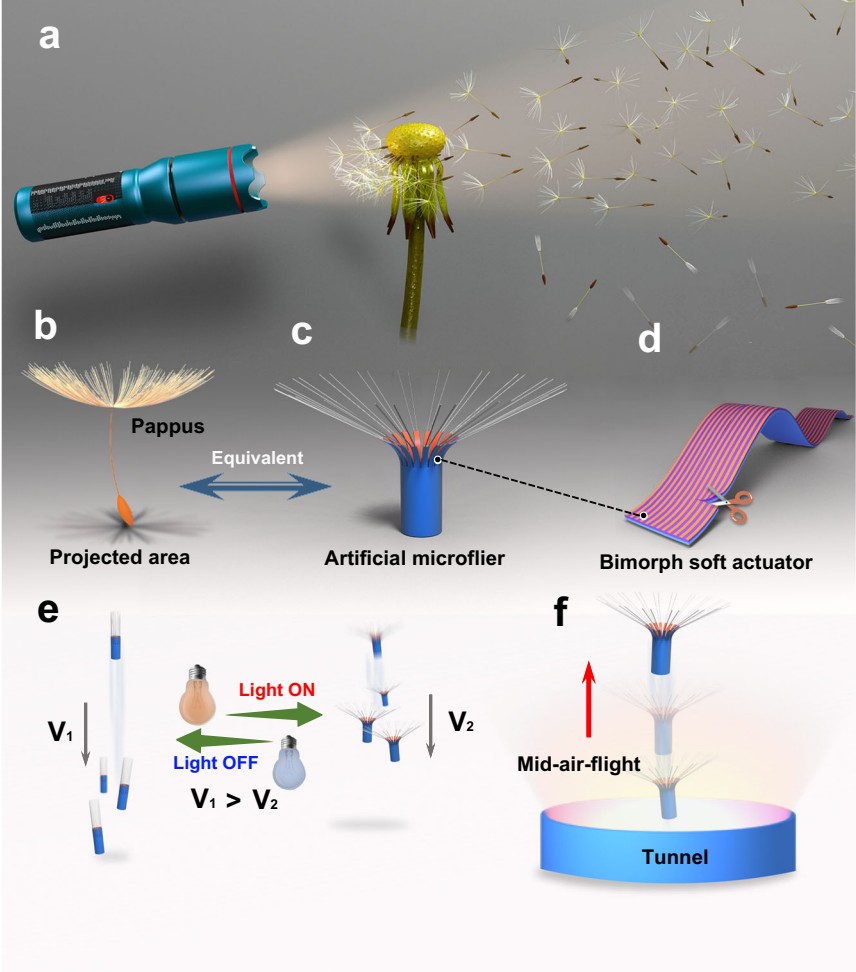

**Fig. 1 | Light-fueled dandelion-inspired artificial microflier and its internal structure. a** Schematic illustration of dandelion-inspired microflier in the air upon opening and closing its "pappus" in response to light irradiation. **b** A schematic dandelion seed in nature. **c** Structure of dandelion-inspired microflier constructed from (**d**) a bimorph soft actuator film. **e** Light-controlled falling velocity of the microflier. **f** Light-powered mid-air flight of the microflier.

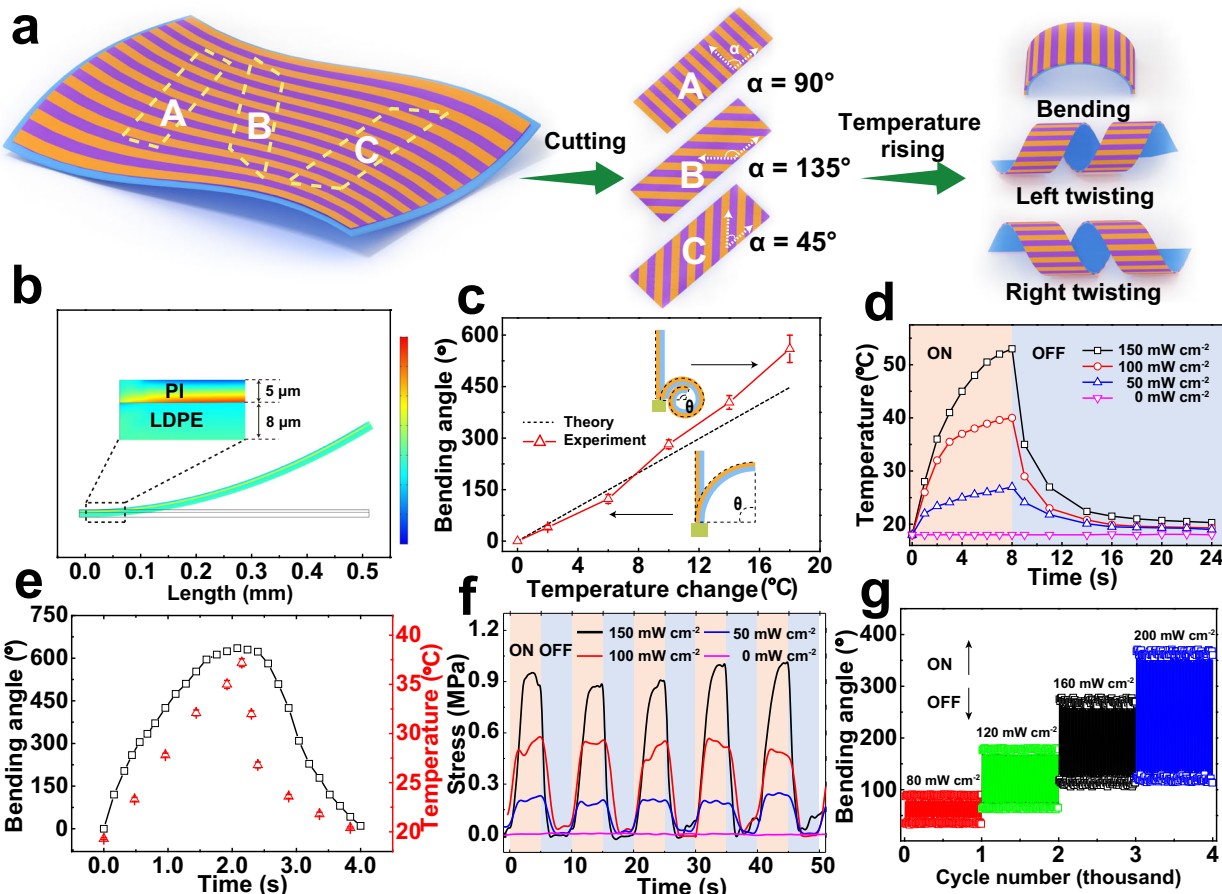

**Fig. 2 | Ultralight and super-sensitive bimorph soft actuator films in response to temperature and light. a** Schematic shape programmability of a bimorph soft actuator film with different cutting angles $\alpha$. **b** COMSOL simulation of the deformation and shear force of the bimorph soft actuator film when exposed to temperature changes ($\Delta T = 16\,°C$). The inset shows the distribution of internal stress at the interface. The color bar corresponds to the color distribution of force magnitudes, where the red areas represent tensile force, whereas blue areas indicate the force with the opposite direction to tensile force. **c** Experimental data and theoretical analysis of the bending angle as a function of the temperature change. The inset chart depicts the bending angle for the bimorph soft actuator film. When the bending angle does not exceed 360°, it is "$\theta$"; When the bending angle exceeds 360°, it is "($\theta + 360°$)". **d** Temperature variation under different light intensities when NIR light is turned on for 8 s and off for 16 s. **e** The bending angle and temperature change of light-driven bimorph soft actuator as a function of time (NIR, 808 nm, 150 mW cm$^{-2}$). The length and width of actuator film are 15 and 3 mm, respectively. **f** Actuation stress of light-driven bimorph soft actuator as a function of time under different NIR light intensities. **g** Repeated actuation of the bimorph soft actuator film under different NIR irradiation intensities. The frequency is 0.5 Hz and irradiation time is 1 s in every cycle.

maximum flight height of ~350 mm (Fig. 1f). Unexpectedly, the microfliers are found to exhibit light-driven upward flight accompanied by autorotating motion, and the rotation mode can be customized in either a clockwise or counterclockwise direction by engineering the shape programmability of bimorph soft actuator films. The research disclosed herein can open up a new avenue toward the development of energy-efficient, and untethered bioinspired artificial microfliers for diverse emerging applications from environmental monitoring to wireless communication.

## Results

### Ultralight and super-sensitive bimorph soft actuator
According to the aerodynamic fundamentals of flying dandelion seed in nature, the size, shape, and mass of the microflier are decisive factors that promote high drag force and low terminal velocities[20–22]. To this end, an ultralight and super-sensitive light-driven bimorph soft actuator was designed and fabricated by using PI (CTE: ~20 K$^{-1}$) and LDPE (CTE: ~280 K$^{-1}$) polymers with superior flexibility and large difference in CTEs (Supplementary Table 1)[18]. Robust photoresponse was enabled by appropriately introducing surface-modified ultrasmall AuNRs with selective optical absorption and high photothermal

conversion efficiency (Supplementary Figs. 1–4)[23], and a strong interface between PI and LDPE was built via electrostatically laminating to avoid the delamination upon scissor-cutting and the release of internal stresses during the actuation process. The details about the fabrication process of bimorph soft actuator film are described in Supporting Information (Supplementary Figs. 5 and 6). Briefly, a PI film was positively charged via layer-by-layer self-assembly coating using poly(allylamine hydrochloride) and poly(4-styrenesulfonic acid) polyelectrolyte layers[24,25], and subsequently electrostatically laminated with a negatively charged LDPE film that was evenly spin-coated with an appropriate amount of surface-modified ultrasmall AuNRs. COMSOL software was adopted to simulate the interaction between layers and the internal stress distribution of the bimorph soft actuator (Fig. 2b). The temperature variation can induce different thermal strains in the two layers, resulting in the bending deformation of the bimorph soft actuator film toward the PI layer (Fig. 2c and Supplementary Fig. 7), where the change in curvature as a function of temperature variation indicates the high sensitivity of the bimorph soft actuator. Taking advantage of the anisotropic properties of the LDPE film[26] (Supplementary Fig. 8), shape-programmable bimorph soft actuators with different shape-deformation modes can be facilely

achieved by altering the cutting angle $\alpha$ (we define $\alpha$ as the angle between the longitudinal direction of the film and the axis of the LDPE along the direction of low CTE value) as shown in Fig. 2a. Upon heating, the uneven expansion of the LDPE layer generates a twisting force whose direction closely depends on the cutting angle $\alpha$ (90°, 135°, and 45°), thus leading to shape deformations from an elongated planar shape into a cylindrical and left- or right-twisting shape, respectively (Supplementary Fig. 9). The fast shape-deformation of the bimorph soft actuator, when exposed to human skin temperature, indicates the high sensitivity to slight temperature changes (Supplementary Movie 1). The introduction of ultrasmall AuNRs endows the bimorph soft actuator with untethered and precisely controlled photoactuation thanks to their optical-to-mechanical energy conversion capacity. Upon near-infrared (NIR) light irradiation (150 mW cm$^{-2}$, 808 nm), the temperature of the bimorph soft actuator could rise rapidly from 18 to 54.2 °C within 8 s (Fig. 2d). It was found that the induced temperature of the bimorph soft actuator was also dependent on the intensities of NIR light, and the temperature increases linearly as a function of the NIR light intensity (Supplementary Fig. 10b). Figure 2e shows the photoactuation property of the bimorph soft actuator under NIR light irradiation (808 nm, 150 mW cm$^{-2}$). The bending angle is defined as the out-of-plane shape deformation angle measured from a fixed end to the distal tip segment after actuation. It was found that the bending angle can change from 0° to 180° in 0.33 s, and it would recover back within 1.62 s upon turning off the NIR light. In addition, the maximum bending angle of the resulting soft actuator was highly dependent on the NIR light intensity, and a maximum bending angle of 30° can be still achieved under 808 nm NIR light irradiation with a very low light intensity of 10 mW cm$^{-2}$ (Supplementary Fig. 10c). It should be noted that up to 0.91 MPa of actuation stress could be achieved in the light-driven bimorph soft actuator (Fig. 2f), and no fatigue failure occurred after more than 1000 cycles, regardless of different light intensities or actuation frequencies (Fig. 2g and Supplementary Fig. 11). The high sensitivity, light weight per unit area (about 2 mg cm$^{-2}$), as well as the robust reversible shape-deformation of the bimorph soft actuator, render it an ideal candidate for fabricating light-powered dandelion-inspired artificial microfliers that follow[27–31].

## Light-controlled falling of dandelion-inspired microfliers

In nature, the dandelion seeds (Taraxacum officinale agg.) are known to consist of an achene and a bundle of around 100 bristly filaments called pappus, in which the achene plays a critical role in maintaining the center of gravity during the flight process. Inspired by dandelion seeds, light-driven artificial microfliers with symmetrical three-dimensional geometry are designed and constructed with tubular-shaped bimorph soft actuator films, of which one end was cut into thin strips and attached with proper amount of fiberglass "pappus" (length of 15 mm, diameter of 25 μm, density of 2.4 g cm$^{-3}$, Taishan Fiberglass, China) (Supplementary Fig. 12). The tubular-shaped bimorph soft actuator with radially symmetrical geometry can function as the "achene" of resulting microfliers to maintain the gravity balance during the flight experiment. The detailed fabrication process of the microfliers is schematically illustrated in Supplementary Fig. 13. Interestingly, the dandelion-inspired artificial microfliers were found to exhibit adaptive light-controlled falling ability in the experiment (Fig. 3a). Here, a commercially available infrared lamp (Philips infrared lamp, 0–350 W) with a wide-band spectrum ranging from 420 to 1000 nm has been used to control the falling of microfliers, as shown in Supplementary Fig. 14. When there was no light irradiation, a microflier with "pappus" at the closed state spent only 0.17 s to fall from a height of 20 to 3.4 cm and the terminal falling velocity was about 0.98 m s$^{-1}$ (Fig. 3b); When light irradiation (50 mW cm$^{-2}$) was turned on, the "pappus" would immediately transform from the closed state to an open state, and the microflier showed a longer falling time of 0.43 s and a much slower terminal falling velocity of ~0.41 m s$^{-1}$ (Fig. 3c). In

contrast, the microflier without "pappus" exhibited a shorter falling time of 0.30 s under similar light irradiation, and terminal falling velocity was ~0.57 m s$^{-1}$ (Fig. 3d, e). The relationship of height and falling time for the microfliers with and without "pappus" is shown in Supplementary Fig. 15 and recorded in Supplementary Movie 2. It is worth noting that the terminal falling velocity of the microflier with "pappus" could be continuously manipulated by changing the light intensity as shown in Fig. 3f, since the opening angle $\phi$ of the "pappus" can increase from 4° to 92° upon changing the light intensity from 0 to 60 mW cm$^{-2}$ (Supplementary Figs. 16 and 17). The resulting microflier was also found to quickly open its "pappus" from a closed state to an open state within ~2.5 s upon being exposed to natural sunlight (~80 mW cm$^{-2}$) at room temperature, and vice versa (Supplementary Fig. 18). Moreover, it was found that an appropriate increase in the attached number of fiberglass strands of the "pappus" could also help to decrease the terminal falling velocity (Supplementary Fig.19). According to the literature[16], drag-enhancing pappus of a dandelion seed can be simplified into a flat porous disk whose projected area is proportional to the drag force. Since the microflier is made from the bimorph soft actuator film, light-driven reversible deformation of its tiny strips would enable the reversible opening/closing of "pappus", thus resulting in tailorable and controllable projected areas (Fig. 3a). When dandelion-inspired microflier falls from a fixed height, the drag force can be expressed as follows[32–34]:

$$F_D = 0.5 C_D \rho A v^2. \tag{1}$$

Where $F_D$ is the air drag force, $C_D$ is the drag coefficient related to the projected area $A$, $\rho$ is the air density (1.2 kg m$^{-3}$), and $v$ is the falling velocity. Thus, the drag force will increase with the increase in the falling velocity until balancing with gravity force, where the falling velocity reaches an equilibrium state and the drag force can be expressed as follows (Supplementary Fig. 20):

$$F_D = mg. \tag{2}$$

Then the above terminal velocity in the balance is rewritten as:

$$v = \left( \frac{2mg}{C_D \rho A} \right)^{1/2}. \tag{3}$$

While the projected area $A$ is affected by the light-triggered opening degree of the "pappus", which in turn allows the superior control of the falling velocity.

Moreover, the terminal falling velocity of as-proposed artificial microfliers with different payload weights was investigated by adding additional payload (from 0 to 24 mg) to the system. As shown in Fig. 3g, the terminal falling velocity of the resulting microflier with its "pappus" at both open and closed states is found to increase when the additional payload weight is increased from 0 to 24 mg, but the terminal falling velocity at the open state is lower (Philips infrared lamp, 50 mW cm$^{-2}$). Interestingly, the opening angle of the "pappus" of the resulting microflier at the open state is found to decrease when the additional payload weight is increased, while microflier without light irradiation always keeps at the close state no matter how much payload it carries. According to Eq. (3), it can be inferred that the increase in the terminal falling velocity of the microfliers with different payloads might result from the increase of weight $m$ and decrease of projected area $A$. However, when the weight of the payload is too heavy (>25 mg) in the experiment, there is no obvious difference in terminal falling velocity for the resulting microflier with its "pappus" at both opening and closed states.

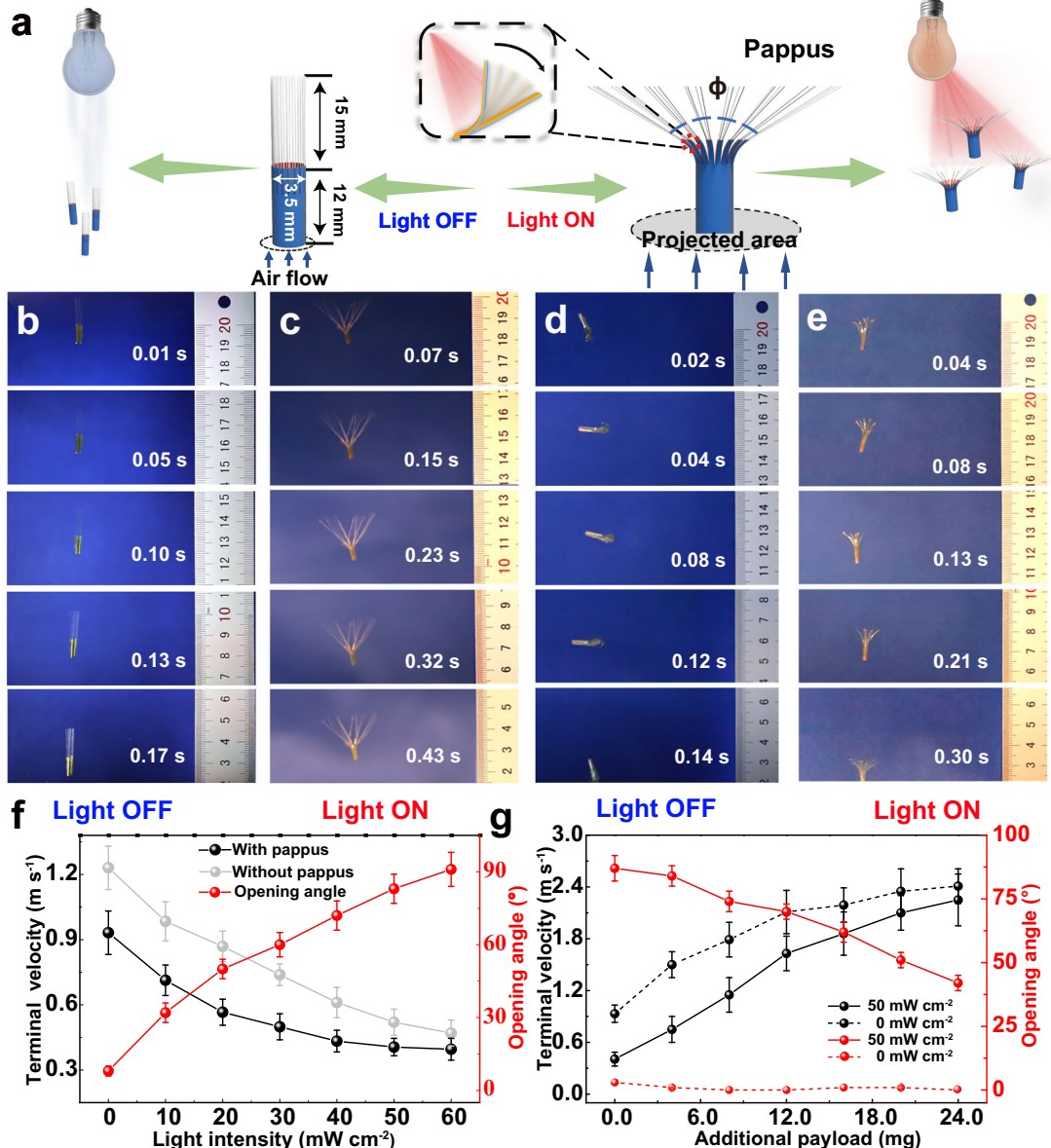

**Fig. 3 | Light-controlled falling velocity of dandelion-inspired microfliers.**
**a** Schematic illustration of the light-controlled falling mechanism of the microflier by which the projected area of the "pappus" affects the drag force and the falling velocity accordingly. Photographs showing the time and falling height of the microfliers with "pappus" when the light irradiation is (**b**) turned off and (**c**) turned on, and the microfliers without "pappus" when the light irradiation is (**d**) turned off and (**e**) turned on. **f** The opening angle φ of "pappus" and the terminal falling velocity of microfliers with and without "pappus" as a function of the light intensity. **g** The terminal falling velocity and opening angle of artificial microflier with different payload under light irradiation (Philips infrared lamp, wavelength range: 420–1000 nm, 50 mW cm⁻²) during the falling process.

## Light-fueled mid-air flight of dandelion-inspired microfliers

To achieve light-fueled mid-air flight of the dandelion-inspired artificial microfliers, we judiciously constructed an optical setup composed of an infrared light bulb (Philips infrared lamp, 0–350 W, wavelength range: 420–1000 mm) and a tunnel that helped to generate a vertical updraft and enhance the light intensity in a confined space, as shown in Fig. 4a and Supplementary Figs. 21 and 22. In the experiment, it was found that the updraft velocity is highly dependent on the power of light, the position of the light source, and the size of the tunnel. Therefore, we fixed the size of the tunnel (40 cm in length and 10 cm in diameter) and the position of the light source (located at a distance of 4 cm from the bottom of the tunnel). When the microflier was held at the height of 5 cm above the center of the tunnel's outlet with a tweezer, it rapidly transformed from a closed to an open state upon light irradiation as depicted in Fig. 4b, d. Upon released from the

tweezer, the microflier was found to be able to fly away immediately and smoothly, and an average flying velocity of 72 mm s⁻¹ and flight height of 274 mm were observed (Fig. 4c, e). The full motion trajectory of artificial microfliers is shown the Fig. 4f and Supplementary Movie 3. It should be noted that the microfliers that cannot open their "pappus" (always at a closed state) are unable to exhibit mid-air flight under similar light irradiation, which indicates that light-driven dandelion-like morphing from a closed state to an open state is of vital importance for enlightening the flight of the microfliers (Supplementary Fig. 23). Moreover, it was found that the flight performance of the microflier can be significantly influenced by the release height above the tunnel's outlet, which might result from the difference in light intensity and "pappus" opening angle of artificial microfliers at different heights (Supplementary Fig. 24). For example, when the microflier is released close to the tunnel's outlet (release height

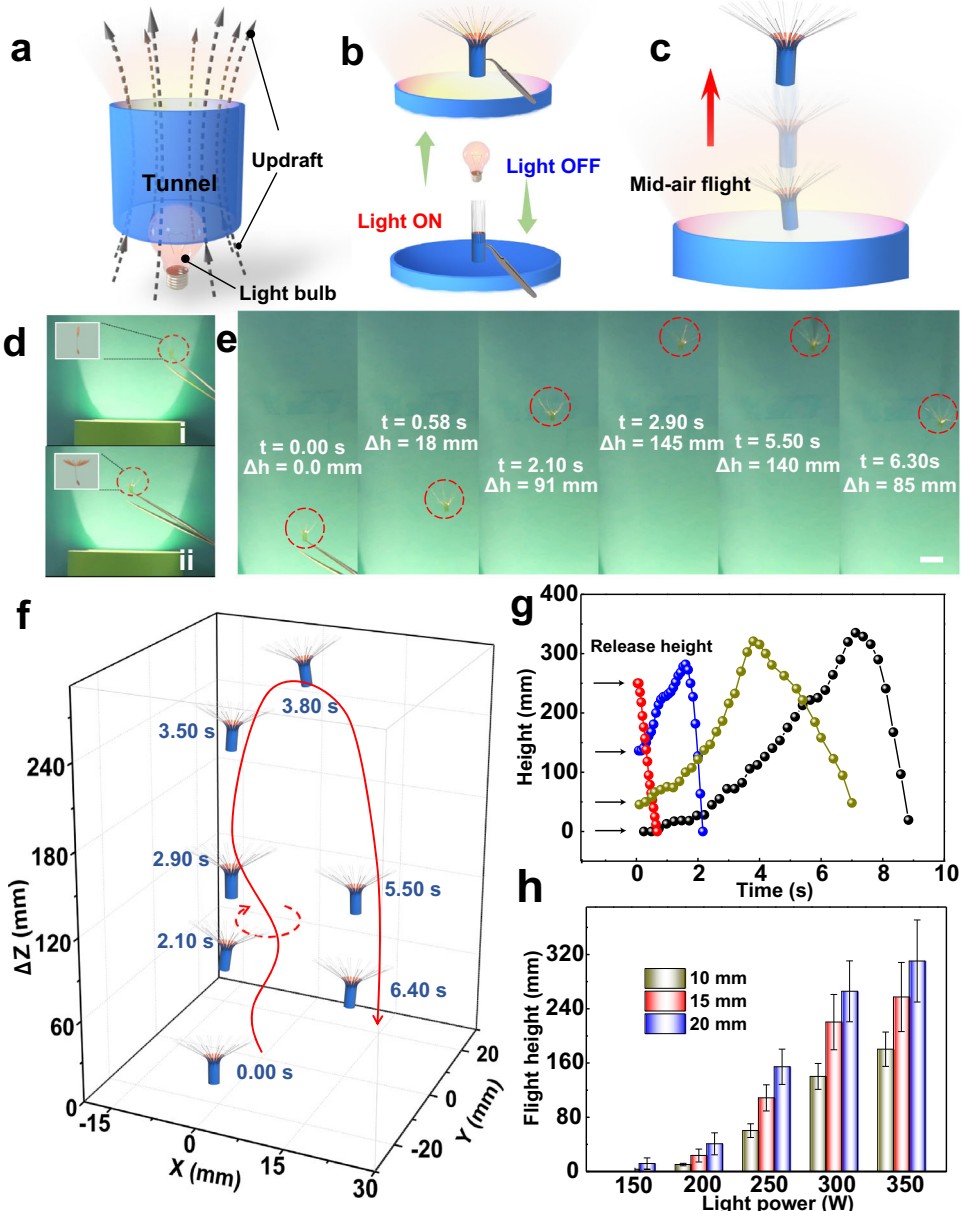

**Fig. 4 | Light-powered mid-air flight of dandelion-inspired artificial microfliers and visualization of the air flow behavior.** Schematic illustration of (**a**) the optical setup for light-powered mid-air flight, **b** reversible opening and closing of the "pappus" of the microflier above the tunnel upon light irradiation and **c** the upward flight of the microflier driven by a light-induced updraft. **d** Images of the microflier with the "pappus" at (I) the closed state and (II) the open state in response to light irradiation (300 W). **e** Image sequence of the microflier flight with (**f**) the corresponding tracking of its trajectory (scale bar: 20 mm). **g** The flight process of microflier (the length of "pappus" is 15 mm) driven by light (300 W) upon released from different heights above the tunnel's outlet. **h** The average flight altitude of the microfliers with different "pappus" lengths (10, 15, and 20 mm) at different light powers ranging from 150 to 350 W (release height of 50 mm).

~0 mm), the maximum flight time and height could reach at ~8.9 s and ~350 mm, respectively. Upon increasing the release height, the flight failure would largely increase until the microflier cannot fly at a release height of 250 mm (Fig. 4g). Interestingly, the flight height of the microflier could also be controlled by adjusting the light power and the length of "pappus" (Fig. 4h). Under constant light irradiation (Philips infrared lamp, 300 W), the flight height of microfliers was found to increase with an increase in the number of fiberglass strands, because an increased number of fiberglass could greatly increase the projected area and thus enhancing the lift force (Supplementary Fig. 25). In the experiment, we set the number of fiberglass at ~40 strands since a further increase of the number of fiberglass strands can not bring a

significant enhancement of flight height, and it is also challenging to technically integrate too many fiberglass strands on such a microflier. Importantly, it was found that light-fueled mid-air flight of microfliers can be even achieved under a simulated solar light (Ceaulight lamp, 300 W, wavelength range: 350–1000 nm), and a sustained flight time of ~2.4 s and a maximum flight height of ~160 mm were observed, as shown in Supplementary Fig. 26.

To further examine the flight mechanism by which light drives the flight of the microflier, flow visualization experiments were carried out to directly observe the flow behavior above the tunnel[16,33]. Unlike the only Brownian motion of the air that occurred when the light was turned off, it was found that an evident updraft above the tunnel was

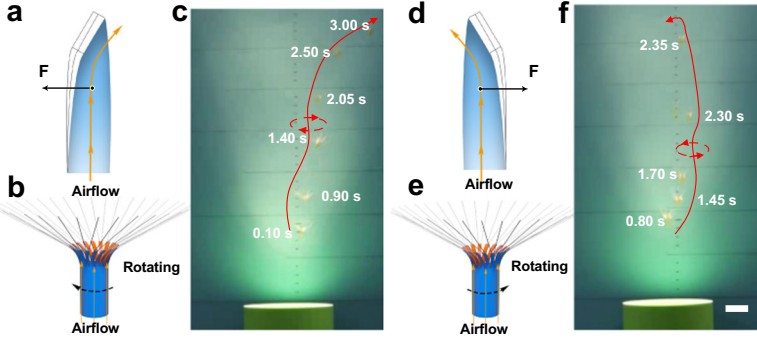

**Fig. 5 | Autorotation flight of dandelion-inspired artificial microfliers.**
**a** Schematic of the horizontal force generated by the airflow around the bimorph thin strips with right-handed twisting; **b**, **c** Clockwise rotation of the corresponding microflier during the flight. **d** Schematic of the horizontal force generated by the airflow around the bimorph thin strips with left-handed twisting; **e**, **f** Counterclockwise rotation of the corresponding microflier during the flight (scale bar: 25 mm).

generated due to the light-induced heating of the surrounding air. Light-induced updraft velocity from 0 to 1.25 m s⁻¹ can be easily obtained by changing the power of light source (Philips infrared lamp, 0–350 W) (Supplementary Fig. 27 and Supplementary Movie 4). To further observe the flow distribution around the microfliers and how updraft would be interacting with thick boundary layers surrounding each fiberglass strand, we used same-sized microfliers with different fixed "pappus" opening angles of 0°, 60°, 90°, and 120° to study their interaction with surrounding airflow. Interestingly, a separated vortex ring was observed above the microflier, and the size of the vortex rings is found to increase with an increase in the "pappus" opening angle, and the observation of a separated vortex ring is very clear above the microflier with a "pappus" opening angle of 120° as shown in Supplementary Fig. 28 and Supplementary Movie 5. Moreover, there is no significant difference in the size and shape of air vortex ring for the microfliers with blocked tube and open tube, which indicates that the airflow should be more acting on the pappus and limited amount of airflow passing through the tube of the microflier (Supplementary Fig. 29). Similar to that observed above the natural dandelion seeds[16], a reduced pressure could be generated above the "pappus", which would provide an effective drag enhancement to balance the gravitational force and keep the microflier aloft accordingly (Supplementary Fig. 30).

## Programmable autorotation flight of artificial microfliers

Unexpectedly, the flight of the microfliers was found to be accompanied by rotation during the flight, such autorotation was found to be either clockwise or counterclockwise, and the probability of occurrence for each was the same through statistical analysis of a large number of flight experiments. This intriguing phenomenon might result from the shape programmability of the bimorph soft actuator films. During the process of fabricating the dandelion-inspired artificial microfliers, one end of as-prepared tubular-shaped bimorph soft actuator film needs to be manually cut into many thin strips, and it is difficult to precisely and accurately cut strips with right angles so that angular deviations by over- or under-cutting result in slightly left-handed or right-handed twisting of the thin strips (Supplementary Fig. 31). As a result, these microfliers can generate torque when encountering the updraft, thus producing the rotation phenomenon during the flight of the microfliers. To confirm the above assumptions, dandelion-inspired artificial microfliers with both right-handed and left-handed twisted thin strips were specifically designed and fabricated. Interestingly, clockwise rotation during the flight was observed in the microfliers with right-handed twisted thin strips, whereas counterclockwise rotation during the flight was observed in the microfliers with left-handed twisted thin strips as shown in Fig. 5 and

Supplementary Movie 6. In this way, the flight mode of the microflier can be facilely manipulated by controlling the shape programmability of the advanced actuator materials, which is of paramount significance for the development of new-generation insect-scale untethered MAVs with programmable controllability and energetically efficient heavier-than-air flight.

As it is well known, the aerodynamic flight of microfliers is one of the most complex modes of locomotion, and the conservation of dynamic balance during locomotion is of particular importance in keeping steady flight[3,32], which mainly depends on how to effectively deal with aerodynamic interaction between microflier and airflow. Recently, a dandelion-inspired artificial disperser with a two-dimensional structural symmetry was reported based on the rectangular light-responsive liquid crystalline elastomer films attached with fabric filaments, where the lift-off action was observed under the combined assistance of wind flow and light irradiation[34]. However, the resulting disperser exhibits some limitations in terms of dispersal performance as a consequence of its two-dimensional structural asymmetry. It is a challenging task to align the direction of wind flow and the position of the light source, which might result in vortex instability and significantly reduce drag force per unit area. As a result, the artificial disperser showed a flight height of less than 25 mm and a short flight time of <1 s under a significant amount of incident power density of 280 mW cm⁻². Here, we believe that superior light-driven mid-air flight performance of our microfliers benefits from the following two aspects: (1) the as-proposed dandelion-inspired artificial microfliers exhibit three-dimensional structural symmetry and symmetrical shape-morphing under appropriated light irradiations; (2) the judiciously designed photoactuation setup is able to generate a vertical updraft and spatially symmetrical light field, and it is worth noting that the updraft and the light field are axisymmetric since they are generated from the same light source. When held at the center of the tunnel's outlet, the resulting microflier would symmetrically open its "pappus" and obtain balanced lift force while symmetrically interacting with the vertical updraft. As a result, the microflier is able to achieve a mid-air flight above a light source with a sustained flight time of ~8.9 s and a maximum flight height of ~350 mm in the experiment. In theory, our as-proposed microfliers would also take off inside a wind tunnel while using separately light to control their deformations. In the experiment, however, it was found that the flight failure greatly increased when using a light source and wind tunnel simultaneously, and it should be noted that autorotation flight mode was no longer observed because it is also a challenging task to assure the axisymmetry of the vertical wind and light field through aligning the direction of wind flow and the position of light source. Likewise, real dandelion seed with an open pappus was found to exhibit light-driven mid-air

flight above the light irradiation (Philips infrared lamp, 300 W), but autorotation flight mode was rarely observed in the real dandelion seeds (Supplementary Movie 7 and Supplementary Fig. 32), confirming that shape-programming of soft actuator plays an important role in autorotation motion genesis.

## Discussion

In summary, we report a general strategy to fabricate light-driven dandelion-inspired artificial microfliers composed of an ultralight and super-sensitive tubular-shaped bimorph soft actuator film, of which one end is cut into thin strips and attached with ~40 strands of fiber-glass ("pappus"), and the resulting "pappus" can be reversibly actuated between a closed state and an open state under external light irradiation. The bimorph soft actuator was designed and fabricated by using lightweight PI and LDPE polymers with a large difference in CTEs, and the ultrasensitive photoresponse was enabled by appropriately introducing surface-modified ultrasmall AuNRs with selective optical absorption and high photothermal conversion efficiency, and a robust interface between PI and LDPE was built via electrostatically laminating to avoid the delamination upon scissor-cutting and the release of internal stresses during the actuation process. Like dandelion seeds in nature, the falling speed of artificial microfliers in air can be easily modulated by changing the opening angle of the "pappus" upon light irradiation. The terminal falling velocity of the microflier with opening "pappus" can be reduced to ~0.41 m s$^{-1}$, which is more than half lower than that of the microflier with closed "pappus" (~0.98 m s$^{-1}$). Interestingly, light-driven mid-air flight of dandelion-inspired artificial microfliers was observed under a commercially available infrared lamp and a simulated solar light, which results from the light-driven updraft and the existence of a separated vortex ring above the microflier as confirmed by flow visualization experiment. The flight performance of the microfliers was found to be closely related to diverse factors such as the intensity of light irradiation, the length and number of "pappus", and the release height of microfliers. Importantly, the flight of the microfliers was found to be accompanied by rotation, and the clock-wise or counterclockwise flight mode of the microflier can be precisely manipulated by controlling the shape programmability of the bimorph soft actuator film. The novelty of this work lies in several aspects: (1) the development of ultralight and super-sensitive bimorph soft actuator film that can be arbitrarily scissor-cut and reversibly photo-actuated; (2) the design and fabrication of dandelion-inspired artificial microfliers exhibiting light-controlled falling velocity and light-fueled mid-air flight; (3) direct observation of a separated vortex ring above the microflier by flow visualization experiment; (4) precise control over the flight motion of the microfliers through programmable materials. The concept of dandelion-inspired artificial microflier can be a versatile platform that can be constructed using almost any ultrafast shape-morphing soft materials such as photoresponsive liquid crystal polymers[35–41], and ferromagnetic soft materials[42–44].

Recently, bioinspired artificial microfliers have been demonstrated by using different forms of actuation such as piezoelectric actuators[9], dielectric elastomer actuators[5,45], and electromagnetic actuators[46]. Although these actuators are known to possess large force output and fast response speed, they need to be integrated with complicated electronics or tethered to a power supply system in an onboard or off-board manner, which inevitably leads to some limitations, such as disturbed flight balance, and extra weight decreasing the flight efficiency of microfliers[8,9]. In contrast, untethered artificial microfliers based on emerging photoactuators have many attractive advantages such as wireless actuation with no need for physical connection between the actuators and the heavy batteries, and spatiotemporally selective controllability through changing the physical parameters of light (intensity, wavelength, and polarization). It should be noted that light is one of the most versatile, inexhaustible, and sustainable energy sources in nature, and the future development of artificial microfliers that can be powered by sunlight (<100 mW cm$^{-2}$) could facilitate many unprecedented features such as flight control by natural sunlight, and real-time adaptive flight according to sunlight shadow. Compared to the bioinspired wind-dispersal microfliers with fixed morphology in the literature[12,17], the development of artificial microfliers with sunlight-adaptive morphology should be much desired for diverse practical applications.

However, the artificial microfliers proposed in this research are still at a preliminary stage and some critical issues and limitations need to be addressed: (1) The difference in terminal velocity of microfliers between the open state and the closed state is found to reduce even with a payload of around 24 mg. (2) The current design has limited flight control capability because it does not yet support in-device control of the microflier when changing its state. (3) The lack of an end-to-end system integrated with multifunctional sensors (pressure, temperature, and humidity sensors) and an electronic microcontroller results in a microflier that cannot yet demonstrate flight attitude feedback and environment monitoring[17,47]. Therefore, future research should consider to improve the system in several aspects such as increasing the photoactuation efficiency of soft actuators, enhancing the lift force of microfliers, and further miniaturizing the size and weight of integrated electronic microcomponents. Furthermore, light-fueled mid-air flight with unexpected rotation and programmable control capability might provide some insights into the development of new-generation artificial microfliers and untethered spacecraft that could adaptively travel and operate in space with reliable and long-acting propulsion from electromagnetic radiations, which could find important applications in diverse fields ranging from environmental monitoring and wireless communication to foldable solar sail and advanced robotic spacecraft[48–50].

## Methods

### Fabrication of bimorph soft actuators

To obtain a positively charged surface, the PI film is firstly dipped into the poly(allylamine hydrochloride) (PAH, Aldrich) solution for at least 5 min, then rinsed with deionized water. Immediately thereafter, the PI film is immersed in the poly(4-styrenesulfonic acid) (PSS, Aldrich) solution for 5 min and rinsed in deionized water, so constituting the 1st cycle of the assembly process. At least five cycles are deposited before the last PAH layer deposition. AuNRs inks are prepared by adding some waterborne acrylic adhesive into the solution of surfaced modified ultrasmall AuNRs. The negatively charged LDPE film is fabricated by plasma treatment (PDC-32G-2, China), which is followed by spin-coating of the AuNRs inks. Bimorph soft actuators are fabricated by electrostatically laminating the positively charged PI film and negatively charged LDPE film.

### Fabrication of dandelion-inspired microfliers

To fabricate a dandelion-inspired microflier, one end of the bimorph soft actuator film ($\alpha = 90°$, 14 mm × 12 mm) is cut evenly along the longitudinal direction to obtain many uniform thin strips (5 mm in length and ~0.7 mm in width), and a tubular bimorph soft actuator is further constructed with the LDPE film as the inner layer and the PI film as the outer layer as schematically shown in Supplementary Fig. 13. The artificial microflier can be obtained after the thin strips were attached with an appropriate amount of fiberglass strands with a length of 10–20 mm ("pappus"), and the total weights of all microfliers are adjusted to about ~4 mg in the experiments.

### Light-controlled falling experiments

To avoid disturbance of surrounding air, all the experiments were conducted in a confined space. At the top of the space, an infrared light (Philips infrared lamp, 0–350 W, wavelength range: 420–1000 nm) was set to control the shape-morphing of microfliers, and the microflier was released from a proper height upon light irradiations. The camera

(Canon, EOS 6D Mark II, Japan) was used to record the fall process of the microflier and its falling velocity was analyzed by Corel Video Studio (Corel, X9, Canada) and Photoshop (Adobe, CC 2019, American).

### Light-powered mid-air flight experiments

In the flight experiment, an airflow tunnel (40 cm long, 10 cm diameter) was designed to generate updrafts above an infrared light (Philips infrared lamp, 0–350 W, wavelength range: 420–1000 nm) and a simulated solar light (Ceaulight lamp, 0–300 W, wavelength range: 350–1000 nm). The microflier was held above the tunnel's outlet with a tweezer, and the light-induced updraft provides the microfliers with lift force. All the experiments were conducted in a confined space to prevent surrounding airflow disturbance.

### Flow visualization experiments

The flow visualization experiment was implemented in a confined space to prevent surrounding airflow disturbance. A fog machine (Yakay PT-1500) was used to seed the air through the airflow tunnel with a length of 400 mm and diameter of 100 mm, and it was illuminated by using a 10-W laser LDY DualPower 304 (diode-pumped, dual cavity, Q-switched Nd: YLF) with a wavelength of 532 nm. The videos were obtained using Canon EOS 6D Mark II (for low-speed videos) and 10-bit high-speed CMOS cameras with 200 mm Micro-Nikkor lens (for high-speed videos). Particle image velocimetry (PIV) was performed using a 532 nm laser to illuminate air that is seeded with smoke. The TR-PIV (Time-resolved particle image velocimetry) was employed to measure the flow field of the turbulent boundary layer in the x−y plane (135 mm × 70 mm). According to the flow velocity, the high sampling frequency was set to 400 Hz and 4000 instantaneous snapshots of particle images were then captured. There were about 20 particles in each interrogate window (32 × 32 pixels) and the overlap rate was 75%. Universal outlier detection in Dantec Dynamic Studio was used to remove the noise, and the proportion of interpolation vectors was <1%.

## Data availability

The authors declare that data supporting the findings of this study are available within the paper and its Supplementary Information, and also from the authors upon request.

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

## Acknowledgements
L.W. acknowledges support from the National Natural Science Foundation of China (Nos. 52173181 and 51973155), Tianjin Science Fund for Distinguished Young Scholars (22JCJQJC00060), and Natural Science Foundation of Tianjin City (20JCYBJC00810). W.F. acknowledges support from Key Program of National Natural Science Foundation of China (No. 52130303). L.W. and W.F. acknowledge support from National Key R&D Program of China (2022YFB3805702).

## Author contributions
L.W. and W.F. supervised the research. L.W. conceived the project and designed the experiments. Y.C. and X.Z. performed the experiments. Y.C., C.V., X.Z., X.Y. and L.W. analyzed the data. L.W., C.V., Y.C. and X.Z. wrote the manuscript. All authors discussed the data and contributed to the writing of the manuscript.

## Competing interests
The authors declare no competing interests.
