## [Peer Review File · Nature Communications]

REVIEWER COMMENTS

Reviewer #1 (Remarks to the Author):

This paper demonstrates a very interesting demonstration of a ultralight weight biomorph soft actuator film that is in the shape of a dandelion seed. The authors demonstrate microfliers where the velocity of these devices can be modified based on the light that is incident on these structures.

I think this paper eventually will be interesting to see at Nature Communications. It is significantly pushing the state-of-the-art, in terms of soft actuator design, of what can be done using dandelion-inspired microflier. I think the demonstration would inspire more research in this nascent area and will potentially lead to more practical demonstrations.

That said, I do have several important concerns that the authors should address in a major revision.

1) Comparison with other actuators. While it is interesting that the authors were able to demonstrate a soft actuator, it looks like it does use a significant amount of incident power. The incident power density is around 50-200 mW/cm². That is a lot of energy that is being used. This is clear in Fig. 2d where the temperature of the device itself changes significantly from 20 to 50C. This does beg the question, from a energy consumption perspective, how does this actuation mechanism compare to using piezoactuators or even magnetic-based approaches? Is it more energy efficient to provide the same change in state? I think answering this important to understand if these soft actuators will end up being used in practice since at these tiny sizes, the power consumption is an important variable to consider. Further, the specific actuators to be used also are affected by weight. A piezoactuators might need voltage converters that might be inefficient and weight quite a bit. However, they might not be high energy. Magnetic based actuators are efficient but the weight of the magnets contribute to the overall weight. I think understand this tradeoff for the proposed actuator and where it would stand in comparison across multiple dimensions will help better position this work.

2) I might have missed this but I did not see an evaluation of different payload sizes. A microflier like this will likely have a microcontroller, a solar cell (?) and sensors similar to Iyer etal (Nature'22). These end up weight quite a bit (10-15 mg) if we were to go with what was claimed in prior work. So I am not sure showing the terminal velocity with a payload-free system is meaningful from a practical perspective since it cannot be realized in practice. Further adding payload to the system does increase your terminal velocity. Can the authors provide an analysis of how the two states differ and the terminal velocities for the two states as a function of differeny payload weights?

3) While it might be a lot of ask, given where the authors are currently, I would have liked to see an end-to-end system with sensors, microcontroller and solar power to demonstrate a real-system.

4) It is unclear to me what the application of the microflifer that the authors designed, given the constrained they have imposed. Comparing to the prior dandelion sensor design that is passive, this work requires a very closeby NIR power source to actuate the device and go from one state to the other. In practice, what application can afford to have such a NIR power source nearby that is useful. Even if the authors could get the system running with solar power, it does not look like they can control when they can switch between the two states. So as soon as the system is exposed to sunlight it will switch to the open state, with ability to control when exactly it can switch. I think either addressing this significant constraint or providing a compelling set of applications where the current setup is still useful will help position the paper better.

5) Claim about "light-driven". The title and the abstract claim that this work is driven by light. Further, the illustration in Fig. 1 shows a torchlight, which can mislead a reader into believing that this work has been demonstrated with visible light. However it looks like the evaluation has been demonstrated with NIR light. It is unclear to me if the work can work with visible light given its broader band nature compared to a single frequency 808 nm light source that the authors use. Providing some results for visible light (not narrowband signals) would strengthen this work. Similarly a reader would be curious if this work will operate with sunlight. I think in practice the challenge is to get these systems operating on sunlight in outdoor environments since a NIR high power source is not available in practical deployment and their use ends up making the proposed system a lot of "tethered" to the artificial power source limiting its applications.

6) The current paper does not have an explicit limitations section as part of conclusions which would have significantly helped.

nits

1) It will be nice to have the dimensions of our microflier in the Fig. It was hard to understand for me how much energy is being consumed with the dimensions, i.e., how do I convert the power density of the incident NIR light to the amount of energy incident on the microflier?

2) Like this work, prior work uses NIR light to power a piezoactuator wing based system. Citing it and comparing the energy requirements will be helpful to contextualize the soft actuator.

Liftoff of a 190 mg Laser-Powered Aerial Vehicle: The Lightest Wireless Robot to Fly, ICRA, 2018

<https://ieeexplore.ieee.org/document/8460582>

3) The work "judicious" seems out of place. I would recommend to remove it.

Reviewer #2 (Remarks to the Author):

This paper describes the design, fabrication, and characterization of the untethered dandelion-inspired microflier composed of polyimide (PI) and low-density polyethylene (LDPE) layers spin-coated with gold nanorods. In my opinion, this new type of microflier is expected to have the great potential to be utilized as next-generation electronic devices (e.g. Nature 597, 503, 2021 / Adv. Funct. Mater 28, 1803149, 2018). The authors were inspired by wind-dispersed seeds such as a dandelion to motivate their designs and highlight the importance of the light-driven untethered artificial microfliers. Overall, I think that this work is significant and strongly believe this interdisciplinary approach would help to develop novel strategies in the research field of soft robotics/electronics. My questions I would like to mention are below.

(1) A real dandelion dispersal unit consists of pappus disk, pappus beak, and achene. Among them, the achene plays the role of maintaining the center of gravity when dandelion seeds are blown by the wind. I would like to recommend authors to add information about their microflier components equivalent to the achene.

(2) In many parts of the manuscript, schematic illustrations for “(A) real dandelion seeds (Figure 1b)” and “(B) their microfliers (Figure 1c)” are used without distinction, so it may be difficult for readers to understand the manuscript. For example, (B) should be used instead of (A) in Figure 1e&f, 3f, 4b,c,f.

(3) Similarly, the schematic illustration of Figure 3f will need to be improved. There is no distinction between the presence and absence of pappus.

(4) I would like to recommend authors to add information (e.g. diameter, density, and manufacturer) about the fiberglass used as the pappus.

(5) How does the stiffness value and number of fiberglass (“pappus”) affect the flight of the microflier? How would the flight of the microflier change if the fiberglass become curved instead of straight?

Reviewer #3 (Remarks to the Author):

Y. Chen et al. developed a dandelion-inspired microflier that can realize the flight via a light-driven tubular structure. The design is exquisite. Many high-quality articles have demonstrated the flight principle of the dandelion and developed a variety of bionic flight devices, but this article has demonstrated its uniqueness, especially the design of the adjustable rotation direction. I appreciate the novelty of using light responsive polymer for realizing a passive flight with assistance of wind flow. However, there are still many issues that would mislead or cause serious confusion. They relate to experimental design, description, presentation methods, etc, as listed below.

Main issues, the authors should not get around them:

1. The authors declare that one of the novelties for this research is direct observation of a separated vortex ring above the microflier by flow visualization experiment. However, in Movie S5 and Figure S19, no separated vortex ring appeared (there is just some air fluctuation). There is a unique definition about SVR, Nature 562, 414-418 (2018), by using stagnation point. Without such PIV characterization, one cannot say the SRV exists.

2. The flier opens and takes off inside the wind tunnel, this is fine. The heat lamp induces convection and used as a wind tunnel to blow up the structure. This is very confusing. As the very hot lamp functions as a wind, at the same time the light triggers shape morphing. I wonder why the authors

should use hot lamp, since there is no way to repeat the same experiments by other researchers – changing another lamp, completely different convection.

3. Whether a real dandelion can fly up the built-up light-driven channel? This controlled experiment will help a lot to analyze the flight mechanism of this aircraft more.

4. In Movie S5, the open angle of the pappus is different. Is the power of the lamp different? Why is a tunnel needed? If there is not this tunnel, can the microflier fly above the lamp?

5. Why did the lighter flier without pappus in Fig. 3d show a faster falling speed than the flier with pappus in Fig. 3a?

6. As the microflier flies, it leaves the lamp far away. The change of opening angle of the pappus should be demonstrated. The intensity distribution of the lamp needs to be tested. The exact positional relationship of lamp and tube also needs to be described.

7. In all the Figures, the airflow seems to pass through the middle of the tube of the flier. The airflow should be more acting on the pappus.

8. No inset chart is found in Fig.2e. Under the same light intensity, why the temperature curves and values in Fig. 2e and Fig. 2d are different?

9. In Fig. 2b, the description of color bar is missed.

In Movie S2, the fall process was too fast to be seen clearly.

In Fig. 3f, the position of the flier should also be marked in the chart by dots.

The experiment is done with microflier instead of dandelions. It is not proper to use the schematic pictures of dandelions to replace the microflier throughout many Figures in the main text.

The light-driven deformation in Fig. 3a should correspond to the expanded pappus on the right, not the left.

In the cycle experiment as shown in Fig. 2g, it seems intuitively that each set of data does not reach 1000 times. It is recommended to display all the cycle data in some separated wide graphs and place it in supporting information to show more data details.

Our point-by-point response to the respected referees' comments and the changes made to the revised manuscript are as follows.

Reviewer #1:

“This paper demonstrates a very interesting demonstration of an ultralight weight bimorph soft actuator film that is in the shape of a dandelion seed. The authors demonstrate microfliers where the velocity of these devices can be modified based on the light that is incident on these structures. I think this paper eventually will be interesting to see at Nature Communications. It is significantly pushing the state-of-the-art, in terms of soft actuator design, of what can be done using dandelion-inspired microflier. I think the demonstration would inspire more research in this nascent area and will potentially lead to more practical demonstrations. That said, I do have several important concerns that the authors should address in a major revision.”

Our response: We greatly appreciate the respected reviewer for valuable suggestions and kindly recommending the manuscript to be published in Nature Communications. In this work, we take aerodynamic lessons from drag-enhancing parachute flight in nature, and report the design and fabrication of a dandelion-inspired artificial microflier that is composed of an ultralight and super-sensitive tubular-shaped bimorph soft actuator film, of which one end is cut into thin strips and attached with ~40 strands of fiberglass (“pappus”). Interestingly, like dandelion seeds in nature, the falling velocity of as-proposed artificial microflier in air can be facilely controlled by tailoring the degree of deformation of the “pappus” under light irradiation with different intensities.

Importantly, the judiciously designed microflier is able to achieve a mid-air flight above a light source with a sustained flight time of ~ 8.9 s and a maximum flight height of ~ 350 mm thanks to the unique dandelion-like 3D structures that facilitate the formation of a separated vortex ring. Unexpectedly, the resulting microflier is found to exhibit light-driven upward flight accompanied by autorotating motion, and the rotation mode can be customized in either a clockwise or counterclockwise direction by engineering the shape programmability of bimorph soft actuator films. This research can shine light on the development of untethered artificial aerial vehicles with programmable controllability, which could find promising applications in diverse fields ranging from environmental monitoring and wireless communication to foldable solar sail and advanced robotic spacecraft. We sincerely believe that the publication of this work in *Nature Communications* would have a broad impact on a wide range of audiences.

“1) Comparison with other actuators. While it is interesting that the authors were able to demonstrate a soft actuator, it looks like it does use a significant amount of incident power. The incident power density is around 50-200 mW/cm². That is a lot of energy that is being used. This is clear in Fig. 2d where the temperature of the device itself changes significantly from 20 to 50 °C. This does beg the question, from an energy consumption perspective, how does this actuation mechanism compare to using piezoactuators or even magnetic-based approaches? Is it more energy efficient to provide the same change in state? I think answering this is important to understand if these soft actuators will end up being used in practice since at these tiny sizes, the power consumption is an important variable to consider. Further, the specific actuators to be used also are affected by weight. A piezoactuators might need voltage converters that might be inefficient and weight quite a bit. However, they might not be high energy. Magnetic based actuators are efficient but the weight of the magnets contributes to the overall weight. I think understand this tradeoff for the proposed actuator and where it would stand in comparison across multiple dimensions will help better position this work.”

Our response: Many thanks for kind comments. Recently, bioinspired artificial microfliers have been demonstrated by using different forms of actuation such as piezoelectric actuators (Nature, 2019, 570, 491), dielectric elastomer actuators (Nature, 2019, 575, 324; Adv. Mater., 2022, 34, 2106757) and electromagnetic actuators (IEEE Trans. Robot., 2016, 32, 1285). Although these actuators are known to possess large force output and fast response speed, they need to be integrated with complicated electronics or tethered to a power supply system in an onboard or off-board manner, which inevitably leads to some limitations, such as disturbed flight balance, and extra weight decreasing the flight efficiency of microfliers. Compared with piezoelectric and electromagnetic actuators, the optical-to-mechanical energy conversion efficiency of photoactuators or light-driven soft actuators is relatively low according to the literature (Adv. Mater., 2017, 29, 1603483; Nat. Rev. Mater., 2022, 7, 235). However, untethered artificial microfliers based on photoactuators have many attractive advantages such as wireless actuation with no need for physical connection between the actuators and the heavy batteries, and spatiotemporally selective controllability through changing the

physical parameters of light (intensity, wavelength, and polarization). It should be noted that light is one of the most versatile, inexhaustible, and sustainable energy sources in nature, and the future development of artificial microfliers that can be powered by sunlight ($< 100 \text{ mW cm}^{-2}$) could facilitate many unprecedented features such as flight control by natural sunlight, and real-time adaptive flight according to sunlight shadow.

In this work, we design and fabricate an ultralight and super-sensitive tubular-shaped bimorph soft actuator film for dandelion-inspired artificial microfliers. The light-driven bimorph soft actuator was fabricated by using polyimide (PI, CTE: $\sim 20 \text{ K}^{-1}$) and low-density polyethylene (LDPE, CTE: $\sim 280 \text{ K}^{-1}$) polymers with a large difference in CTEs. Robust photoresponse was enabled by appropriately introducing surface-modified ultrasmall AuNRs with high photothermal conversion efficiency, and a strong interface between PI and LDPE was built via electrostatic lamination to avoid the delamination upon scissor-cutting and the release of internal stresses during the actuation process. In the revised manuscript, we further characterized the photoactuation performance of the as-proposed bimorph soft actuator under 808 nm NIR light irradiation with different intensities (10 mW cm^{-2} – 200 mW cm^{-2}). As shown in **Supplementary Fig. 10c** of revised supplemental information, the maximum bending angle of the resulting soft actuator is highly dependent on the NIR light intensity, and a maximum bending angle of 30° can be still achieved under 808 nm NIR light irradiation with a very low light intensity of 10 mW cm^{-2} , which can be attributed to the high photothermal conversion efficiency of embedded ultrasmall AuNRs. Interestingly, the microflier based on the resulting soft actuator film can adaptively change the opening angle of the “pappus” from 4° to 92° upon varying light intensity from 0 to 60 mW cm^{-2} under an infrared lamp (Philips infrared lamp, wavelength range: 420 nm–1000 nm), as shown in **Supplementary Fig. 15** of revised supplemental information. It should be noted that the resulting microflier was also found to quickly open its “pappus” from a closed state to an open state upon being exposed to natural sunlight ($\sim 80 \text{ mW cm}^{-2}$) at room temperature, and vice versa, as shown in **Supplementary Fig. 17** of revised supplemental information. Fig. 2d indicates the temperature variation of bimorph soft actuator film under different light intensities when 808 nm NIR light is turned on for 8 s and off for 16 s, which can help to verify the high photothermal conversion efficiency of ultrasmall AuNRs. In fact, the bimorph soft actuators are found to be highly sensitive to temperature variation, and fast and reversible actuation can be even achieved upon being exposed to human skin temperature as shown in **Supplementary Movie 1**. To the best of our knowledge, as-proposed light-driven bimorph soft actuators outperform most of those reported in the literature in terms of response time and deformation curvature (Mater. Horiz., 2021, 8, 728; Adv. Mater., 2018, 30, 1703554).

Per suggestion, **Supplementary Fig. 10c**, **Fig. 15**, and **Fig. 17** have been added to the revised supplemental information, and the corresponding descriptions have been added in the revised manuscript accordingly. The tradeoff for the proposed photoactuator compared with other actuators has been discussed in the conclusion. The details are shown as follows:

On page 9 of revised manuscript: “Recently, bioinspired artificial microfliers have been demonstrated by using different forms of actuation such as piezoelectric actuators, dielectric elastomer actuators, and electromagnetic actuators. Although these actuators are known to possess large force output and fast response speed, they need to be integrated with complicated electronics or tethered to a power supply system in an onboard or off-board manner, which inevitably leads to some limitations, such as disturbed flight balance and extra weight, decreasing the flight efficiency of microfliers. In contrast, untethered artificial microfliers based on emerging photoactuators have many attractive advantages such as wireless actuation with no need for physical connection between the actuators and the heavy batteries, and spatiotemporally selective controllability through changing the physical parameters of light (intensity, wavelength, and polarization). It should be noted that light is one of the most versatile, inexhaustible, and sustainable energy sources in nature, and the future development of artificial microfliers that can be powered by sunlight ($< 100 \text{ mW cm}^{-2}$) could facilitate many unprecedented features such as flight control by natural sunlight, and real-time adaptive flight according to sunlight shadow.”

On page 4 of revised manuscript: “The opening angle θ of the “pappus” can increase from 4° to 92° upon changing the light intensity from 0 to 60 mW cm^{-2} (Supplementary Fig. 15,16). The resulting microflier was also found to quickly open its “pappus” from a closed state to an open state upon being exposed to natural sunlight ($\sim 80 \text{ mW cm}^{-2}$) at room temperature, and vice versa (Supplementary Fig. 17)”

On page 4 of revised manuscript: “In addition, the maximum bending angle of the resulting soft actuator is highly dependent on the NIR light intensity, and a maximum bending angle of 30° can be still achieved under 808 nm NIR light irradiation with a very low light intensity of 10 mW cm^{-2} (Supplementary Fig. 10c).”

On page 3 of revised manuscript: “Compared to other reported soft actuators, the as-proposed bimorph soft actuator outperforms in several aspects, including excellent deformation derived from its large mismatch in coefficients of thermal expansion (CTEs), facile shape programmability, and outstanding sensitivity (e.g., hand temperature can cause large deformation).”

Supplementary Fig. 10c. The measured maximum bending angle for bimorph soft actuator as a function of NIR light intensity (808 nm, 10 – 200 mW cm⁻²).

Supplementary Fig. 15. The “pappus” opening of artificial microflyer under light irradiation. **a** The “pappus” opening of artificial microflyer in response to light (Philips infrared lamp, 50 mW cm⁻², wavelength range: 420 nm~1000 nm). **b** The maximum opening angle of the “pappus” of a microflyer under light irradiation with different intensities (scale bar: 10 mm).

Supplementary Fig. 17. The “pappus” opening of artificial microflyer upon being exposed to natural sunlight (~80 mW cm⁻²) at room temperature.

“2) I might have missed this but I did not see an evaluation of different payload sizes. A microflyer like this will likely have a microcontroller, a solar cell (?) and sensors similar to Iyer et. al. (Nature’22). These end up weight quite a bit (10-15 mg) if we were to go with what was claimed in prior work. So, I am not sure showing the terminal velocity with a payload-free system is meaningful from a practical perspective since it

cannot be realized in practice. Further adding payload to the system does increase your terminal velocity. Can the authors provide an analysis of how the two states differ and the terminal velocities for the two states as a function of different payload weights?” **Our response:** Many thanks for kind suggestion. We have tested the terminal falling velocity of as-proposed artificial microfliers (4 mg) with different payload weights by adding additional payload (from 0 mg to 24 mg) to the system. We also measured the opening angle of the “pappus” of resulting microfliers during the falling process. As shown in **Supplementary Fig. 20** in revised supplemental information, when the additional payload weight is increased from 0 mg to 24 mg, the terminal falling velocity of the resulting microflier with its “pappus” at both open and closed states is found to increase, but the terminal falling velocity at the open state is lower (Philips infrared lamp, 50 mW cm⁻²). Interestingly, the opening angle of the “pappus” of the resulting microflier at the open state is found to decrease when the additional payload weight is increased, while microflier without light irradiation always keeps at the close state no matter how much payload it carries. According to Equation (3), it can be inferred that the increase in terminal velocity of the microfliers with different payloads might result from the increase of weight m and decrease of projected area A . However, when the weight of the payload is too heavy (> 25 mg) in the experiment, there is no obvious difference in terminal falling velocity for the resulting microflier with its “pappus” at both open and closed states.

Per suggestion, **Supplementary Fig. 20** has been added to the revised supplemental information, and the corresponding descriptions have been added to the revised manuscript accordingly. The details are shown as follows:

On page 7 of the manuscript: “Moreover, the terminal falling velocity of as-proposed artificial microfliers (4 mg) with different payload weights was measured by adding additional payload (from 0 mg to 24 mg) to the system. As shown in **Supplementary Fig. 20**, the terminal falling velocity of the resulting microflier with its “pappus” at both open and closed states is found to increase when the additional payload weight is increased from 0 mg to 24 mg, but the terminal falling velocity at the open state is lower (Philips infrared lamp, 50 mW cm⁻²). Interestingly, the opening angle of the “pappus” of the resulting microflier at the open state is found to decrease when the additional payload weight is increased, while microflier without light irradiation always keeps at the close state no matter how much payload it carries. According to Equation (3), it can be inferred that the increase in terminal falling velocity of the microfliers with different payloads might result from the increase of weight m and decrease of projected area A . However, when the weight of the payload is too heavy (> 25 mg) in the experiment, there is no obvious difference in terminal falling velocity for the resulting microflier with its “pappus” at both open and closed states.”

Supplementary Fig. 20. The terminal falling velocity and opening angle of artificial microflier with different additional payload weights under light irradiation (Philips infrared lamp, wavelength range: 420 nm~1000 nm, 50 mW cm⁻²) during the falling process. The weight of the artificial microflier is 4 mg.

“3) While it might be a lot of ask, given where the authors are currently, I would have liked to see an end-to-end system with sensors, microcontroller and solar power to demonstrate a real-system.”

Our response: Thanks for kind comments. In this work, we demonstrate a unique light-driven artificial microflier with dandelion-inspired 3D geometry. Light-controlled falling velocity of resulting microfliers and light-fueled mid-air flight accompanied by autorotating motion have been achieved under external light irradiation (Philips infrared lamp, wavelength range: 420 nm~1000 nm). It is worth noting that the unexpected autorotation mode can be customized in either a clockwise or counterclockwise direction by engineering the shape programmability of bimorph soft actuator films. However, the as-proposed artificial microfliers are still in a very preliminary stage and some critical issues need to be addressed such as low payload weight, limited flight controllability, and inadequate flight performance. The development of an end-to-end system integrated with multifunctional sensors and an electronic microcontroller will be of practical significance, but it is a challenging and formidable task at this current stage. According to recent work (Nature, 2022, 603, 427), the total weight of integrated electronics including sensors, microcontroller, etc. is more than 30 mg, and such a payload is too heavy for the as-proposed artificial microflier as we discussed above. Therefore, future research should consider to improve the system in several aspects such as increasing the photoactuation efficiency of soft actuators, enhancing the lift force of microfliers, and further miniaturizing the size and weight of integrated electronic microcomponents. Per suggestion, the following discussions have been added to the conclusion of revised manuscript:

On page 9 of the manuscript: “However, the as-proposed artificial microfliers are still in a very preliminary stage and some critical issues need to be addressed such

as low payload weight, limited flight controllability, and inadequate flight performance. The development of an end-to-end system integrated with multifunctional sensors and an electronic microcontroller will be of practical significance, but it is a challenging and formidable task at this current stage. Therefore, future research should consider to improve the system in several aspects such as increasing the photoactuation efficiency of soft actuators, enhancing the lift force of microfliers, and further miniaturizing the size and weight of integrated electronic microcomponents.”

“4) It is unclear to me what the application of the microflier that the authors designed, given the constrained they have imposed. Comparing to the prior dandelion sensor design that is passive, this work requires a very close by NIR power source to actuate the device and go from one state to the other. In practice, what application can afford to have such a NIR power source nearby that is useful. Even if the authors could get the system running with solar power, it does not look like they can control when they can switch between the two states. So as soon as the system is exposed to sunlight it will switch to the open state, with ability to control when exactly it can switch. I think either addressing this significant constraint or providing a compelling set of applications where the current setup is still useful will help position the paper better.”

Our response: Many thanks for kind comments. It should be pointed out that in our experiment light-controlled falling velocity and light-fueled mid-air flight of as-prepared artificial microfliers were performed under an infrared lamp with a wide spectral range (Philips infrared lamp, wavelength range: 420 nm~1000 nm) as shown in **Supplementary Fig. 14** of revised supplemental information. Interestingly, the resulting microflier was found to quickly open its “pappus” from a closed state to an open state upon being exposed to natural sunlight (~80 mW cm⁻²) at room temperature, and vice versa (**Supplementary Fig. 17**). As shown in **Supplementary Fig. 26** of revised supplemental information, we also found that light-fueled mid-air flight of microfliers can be even achieved under a simulated solar light (Ceaulight lamp, 300 W). Although the as-proposed artificial microfliers are still far from the practical application at this stage, future development of artificial microfliers powered by sunlight could facilitate many unprecedented functions such as flight control by natural sunlight, and real-time adaptive motion according to sunlight shadow or sunlight at different times. Compared to the bioinspired wind-dispersal microfliers with fixed morphology in the literature (Nature, 2022, 603, 427), the development of artificial microfliers with sunlight-adaptive morphology should be much desired for diverse practical applications from environmental monitoring to wireless communication and beyond. Furthermore, light-fueled mid-air flight and programmable unexpected autorotation of flight mode could offer some insights for the development of new-generation insect-scale untethered artificial aerial vehicles with self-regulation and programmable controllability. Per suggestion, the constraint of as-proposed artificial microfliers and potential applications have been discussed in the conclusion of revised manuscript accordingly.

“5) Claim about “light-driven”. The title and the abstract claim that this work is driven by light. Further, the illustration in Fig. 1 shows a torchlight, which can mislead a reader into believing that this work has been demonstrated with visible light. However, it looks like the evaluation has been demonstrated with NIR light. It is unclear to me if the work can work with visible light given its broader band nature compared to a single frequency 808 nm light source that the authors use. Providing some results for visible light (not narrowband signals) would strengthen this work. Similarly, a reader would be curious if this work will operate with sunlight. I think in practice the challenge is to get these systems operating on sunlight in outdoor environments since a NIR high power source is not available in practical deployment and their use ends up making the proposed system a lot of “tethered” to the artificial power source limiting its applications.”

Our response: Thanks for kind comments. When investigating the photoactuation performance of the bimorph soft actuator including speed, deflection range, and stability (Fig. 2), we used an 808 nm NIR light with adjustable power (Changchun New Industries Optoelectronics Technology Co., 0-10W). When investigating the light-controlled falling and light-powered flight performances of as-prepared microflier (Fig. 3, 4, and 5), we used a commercially available infrared lamp with a wide spectral range (Philips infrared lamp, 0-350 W). As shown in **Supplementary Fig. 14** of revised supplemental information, the infrared lamp possesses a wide-band spectrum with a wavelength ranging from 420 nm to 1000 nm. Per suggestion, the photo-mechanical responsive property of as-prepared microfliers has been investigated under natural sunlight. As shown in **Supplementary Fig. 17** of revised supplemental information, the resulting microflier was found to quickly open its “pappus” from a closed state to an open state upon being exposed to natural sunlight ($\sim 80 \text{ mW cm}^{-2}$) at room temperature, and vice versa. As shown in **Supplementary Fig. 26** of revised supplemental information, we also found that light-fueled mid-air flight of microfliers can be even achieved under a simulated solar light (Ceaulight lamp, 300 W, wavelength range: 350 nm-1000 nm). To avoid the influence of air turbulence and unstable temperature in the outdoors, light-fueled mid-air flight of microfliers was also performed under a simulated solar light (Ceaulight lamp, 300 W) in a confined space. It was found that light-fueled mid-air flight of microfliers can be even achieved under a simulated solar light, and a sustained flight time of $\sim 2.4 \text{ s}$ and a maximum flight height of $\sim 160 \text{ mm}$ was observed, as shown in **Supplementary Fig. 26** of revised supplemental information.

Per suggestion, **Supplementary Fig. 14**, **Fig. 17**, and **Fig. 26** have been added to the revised supplemental information, and the corresponding descriptions have been added in the revised manuscript accordingly. The details are shown as follows:

On page 4 of the revised manuscript: *“Here, a commercially available infrared lamp (Philips infrared lamp, 0~350 W) with a wide-band spectrum ranging from 420 nm to 1000 nm has been used to control the falling of microfliers, as shown in Supplementary Fig. 14.”*

On page 7 of the revised manuscript: *“Importantly, it was found that light-fueled mid-air flight of microfliers can be even achieved under a simulated solar light, and a*

sustained flight time of ~ 2.4 s and a maximum flight height of ~ 160 mm was observed, as shown in Supplementary Fig. 26.”

Supplementary Fig. 14. The optical spectra of different light sources used in the experiment: Natural sunlight, simulated solar spectrum (Ceaulight lamp, 0 W~300 W), infrared light (Philips infrared lamp, 0~350 W) and 808 nm NIR light (Changchun New Industries Optoelectronics Technology Co., 0~10 W).

Supplementary Fig. 26. The flight process of as-prepared microflyers above the simulated solar light (Ceaulight lamp, 300 W).

“6) The current paper does not have an explicit limitations section as part of the conclusions which would have significantly helped.”

Our response: Many thanks for kind suggestion. Corresponding discussions have been added in the conclusion of revised manuscript as follows:

On page 9 of the revised manuscript: “Recently, bioinspired artificial microflyers have been demonstrated by using different forms of actuation such as piezoelectric actuators, dielectric elastomer actuators, and electromagnetic actuators. Although

these actuators are known to possess large force output and fast response speed, they need to be integrated with complicated electronics or tethered to a power supply system in an onboard or off-board manner, which inevitably leads to some limitations, such as disturbed flight balance, and extra weight decreasing the flight efficiency of microfliers. In contrast, untethered artificial microfliers based on emerging photoactuators have many attractive advantages such as wireless actuation with no need for physical connection between the actuators and the heavy batteries, and spatiotemporally selective controllability through changing the physical parameters of light (intensity, wavelength, and polarization). It should be noted that light is one of the most versatile, inexhaustible, and sustainable energy sources in nature, and the future development of artificial microfliers that can be powered by sunlight ($< 100 \text{ mW cm}^{-2}$) could facilitate many unprecedented features such as flight control by natural sunlight, and real-time adaptive flight according to sunlight shadow. Compared to the bioinspired wind-dispersal microfliers with fixed morphology in the literature, the development of artificial microfliers with sunlight-adaptive morphology should be much desired for diverse practical applications. However, the proposed artificial microfliers in this research are still in a very preliminary stage and some critical issues need to be addressed such as low payload weight, limited flight controllability, and inadequate flight performance. The development of an end-to-end system integrated with multifunctional sensors and an electronic microcontroller will be of practical significance, but it is a challenging and formidable task at this current stage. Therefore, future research should consider to improve the system in several aspects such as increasing the photoactuation efficiency of soft actuators, enhancing the lift force of microfliers, and further miniaturizing the size and weight of integrated electronic microcomponents. Furthermore, light-fueled mid-air flight with unexpected rotation and programmable control capability might provide some insights into the development of new-generation artificial microfliers and untethered spacecraft that could adaptively travel and operate in space with reliable and long-acting propulsion from electromagnetic radiations, which could find important applications in diverse fields ranging from environmental monitoring and wireless communication to foldable solar sail and advanced robotic spacecraft.”

nits

“1) It will be nice to have the dimensions of our microflier in the Fig. It was hard to understand for me how much energy is being consumed with the dimensions, i.e., how do I convert the power density of the incident NIR light to the amount of energy incident on the microflier?”

Our response: Thanks for kind comments. The dimensions of the as-prepared microflier have been added in Fig. 3a of revised manuscript, and the dimension comparison of a real dandelion seed and as-prepared microflier has been shown in **Supplementary Fig. 12** of revised supplemental information. The consumed energy for the microfliers is difficult to accurately calculate, but the total energy consumption can be roughly estimated as follows. Two types of energy are needed for light-powered flight of microfliers. The first one is optical-to-mechanical energy, which opens the

“pappus” of microfliers to achieve a proper projected area, and the second is the thermal energy transfer originating from light-induced convective updraft, which helps to overcome gravity and makes artificial microfliers lift off and fly in air.

Supplementary Fig. 12. Dimensions comparison of a real dandelion seed and dandelion-inspired microflier. The length of the “pappus” of the microflier can be set to 10 mm, 15 mm, or 20 mm by cutting.

Like this work, prior work uses NIR light to power a piezoactuator wing based system. Citing it and comparing the energy requirements will be helpful to contextualize the soft actuator. *Liftoff of a 190 mg Laser-Powered Aerial Vehicle: The Lightest Wireless Robot to Fly*, ICRA, 2018 <https://ieeexplore.ieee.org/document/8460582>” **Our response:** Per suggestion, the reference has been cited as Ref. 8 in the revised manuscript. It should be pointed out that prior works demonstrate light-powered untethered microfliers based on complicated photovoltaics and integrated electronics. Such previously reported microfliers exhibited a high mass of over 100 mg, a sustained flight time of around 1 s, and a maximum flight height of around 10 cm. In contrast, light-driven dandelion-inspired microfliers in our research are able to achieve a mid-air flight above a light source with a sustained flight time of ~ 8.9 s and a maximum flight height of ~350 mm.

The word "judicious" seems out of place. I would recommend to remove it.”

Our response: Per suggestion. The word "judicious" has been removed in the revised manuscript.

Reviewer #2:

“This paper describes the design, fabrication, and characterization of the untethered dandelion-inspired microflier composed of polyimide (PI) and low-density polyethylene (LDPE) layers spin-coated with gold nanorods. In my opinion, this new type of microflier is expected to have the great potential to be utilized as next-generation

electronic devices (e.g. *Nature* 597, 503, 2021 / *Adv. Funct. Mater* 28, 1803149, 2018). The authors were inspired by wind-dispersed seeds such as a dandelion to motivate their designs and highlight the importance of the light-driven untethered artificial microfliers. Overall, I think that *this work is significant and strongly believe this interdisciplinary approach would help to develop novel strategies in the research field of soft robotics/electronics*. My questions I would like to mention are below.”

Our response: We greatly appreciate the respected reviewer’s highly positive comments on our manuscript. In this work, we report a general strategy to fabricate light-driven dandelion-inspired artificial microfliers composed of an ultralight and super-sensitive tubular-shaped bimorph soft actuator film. The falling velocity of artificial microfliers can be facilely controlled by tailoring the opening degree of the “pappus” with different light intensities. Importantly, light-fueled mid-air flight accompanied by unexpected autorotating motion has been observed in the resulting microfliers. This work might provide some insights into the development of untethered artificial aerial vehicles with programmable controllability, which could find promising applications in diverse fields ranging from environmental monitoring and wireless communication to foldable solar sail and advanced robotic spacecraft. We sincerely believe that the publication of this exciting work on *Nature Communications* would have a broad impact on a wide range of audiences.

(1) *A real dandelion dispersal unit consists of pappus disk, pappus beak, and achene. Among them, the achene plays the role of maintaining the center of gravity when dandelion seeds are blown by the wind. I would like to recommend authors to add information about their microflier components equivalent to the achene.*

Our response: Thanks for kind suggestion. In nature, the dandelion seeds (*Taraxacum officinale* agg.) are known to consist of an achene attached to a bundle of around 100 bristly filaments called pappus, in which the achene plays a critical role in maintaining the center of gravity during the flight process. Inspired by dandelion seeds, we design and fabricate a unique light-driven artificial microflier with symmetrical three-dimensional geometry, which is composed of a tubular-shaped bimorph soft actuator attached with proper amount of fiberglass (“pappus”). The tubular-shaped bimorph soft actuator with radially symmetrical geometry can function as the “achene” of resulting microfliers to maintain the gravity balance during the flight experiment. Accordingly, the following descriptions have been added on Page 4 of revised manuscript:

*“In nature, the dandelion seeds (*Taraxacum officinale* agg.) are known to consist of an achene attached to a bundle of around 100 bristly filaments called pappus, in which the achene plays a critical role in maintaining the center of gravity during the flight process. Inspired by dandelion seeds, light-driven artificial microfliers with symmetrical three-dimensional geometry are designed and constructed with tubular-shaped bimorph soft actuator films, of which one end was cut into thin strips and attached with proper amount of fiberglass “pappus” (length of 15 mm, diameter of 25 μm , density of 2.4 g cm^{-3} , Taishan Fiberglass, China) (Supplementary Fig. 12). The tubular-shaped bimorph soft actuator with radially symmetrical geometry can function*

as the “achene” of resulting microfliers to maintain the gravity balance during the flight experiment.”

(2) In many parts of the manuscript, schematic illustrations for “(A) real dandelion seeds (Figure 1b)” and “(B) their microfliers (Figure 1c)” are used without distinction, so it may be difficult for readers to understand the manuscript. For example, (B) should be used instead of (A) in Figure 1e&f, 3f, 4b, c.f.

Our response: Per suggestion, all the figures have been improved accordingly in the revised manuscript.

(3) Similarly, the schematic illustration of Figure 3f will need to be improved. There is no distinction between the presence and absence of pappus.

Our response: Thanks for kind suggestion. Figure 3f has been improved in the revised manuscript.

(4) I would like to recommend authors to add information (e.g. diameter, density, and manufacturer) about the fiberglass used as the pappus.

Our response: Per suggestion, the information about commercially available fiberglass has been added in the revised manuscript, and supplemental information as follows: “fiberglass (diameter of 25 μm , density of 2.4 g cm^{-3} , Taishan Fiberglass, China).”

(5) How does the stiffness value and number of fiberglass (“pappus”) affect the flight of the microflier? How would the flight of the microflier change if the fiberglass become curved instead of straight?

Our response: Thanks for kind comments. In the revised manuscript, we have investigated the effect of the number of fiberglass strands (the length is fixed at 15 mm) on the flight height, as shown in **Supplementary Fig. 25** of revised supplemental information. Under a constant light irradiation (Philips infrared lamp, 300 W), the flight height of microfliers was found to increase with an increase in the number of fiberglass strands, because an increased number of fiberglass could greatly increase the projected area and thus enhancing the lift force ($F_D = 0.5C_D\rho Av^2$). In the experiment, we set the number of fiberglass at ~40 strands since a further increase of the number of fiberglass strands can not bring a significant enhancement of flight height, and it is also challenging to technically integrate too many fiberglass strands on such a microflier.

The low stiffness value of fiberglass could result in its easy deformation during the falling or flight process, which will reduce the projected area of microfliers and lift force during the experiment. To avoid such limitations, we used fiberglass with high stiffness (~3 GPa) in the fabrication of the microflier. If the fiberglass becomes curved, the vertical projected area could decrease and thus a decreased flight height, which can be further confirmed by the average flight height of the microfliers with different “pappus” lengths (10 mm, 15 mm, and 20 mm), as shown in Fig.5h.

Per suggestion, **Supplementary Fig. 25** has been added to the revised supplemental information, and the corresponding descriptions have been added in the revised manuscript as follows:

On page 7 of revised manuscript: “Under a constant light irradiation (Philips infrared lamp, 300 W), the flight height of microfliers was found to increase with an increase in the number of fiberglass strands, because an increased number of fiberglass could greatly increase the projected area and thus enhancing the lift force (Supplementary Fig. 25). In the experiment, we set the number of fiberglass at ~40 strands since a further increase of the number of fiberglass strands can not bring a significant enhancement of flight height, and it is also challenging to technically integrate too many fiberglass strands on such a microflier.”

Supplementary Fig. 25. The average flight height of the microfliers with different numbers of fiberglass strands (the length of pappus is fixed at 15 mm) under constant light irradiation (Philips infrared lamp, 300 W).

Reviewer #3:

“Y. Chen et al. developed a dandelion-inspired microflier that can realize the flight via a light-driven tubular structure. *The design is exquisite.* Many high-quality articles have demonstrated the flight principle of the dandelion and developed a variety of bionic flight devices, but *this article has demonstrated its uniqueness, especially the design of the adjustable rotation direction.* I appreciate the novelty of using light responsive polymer for realizing a passive flight with assistance of wind flow. However, there are still many issues that would mislead or cause serious confusion. They relate to experimental design, description, presentation methods, etc, as listed below.”

Our response: We greatly appreciate the respected reviewer for his/her encouraging comments on the novelty and significance of our manuscript. In this work, we demonstrate light-driven dandelion-inspired microfliers based on ultralight and super-sensitive tubular-shaped bimorph soft actuator. Like dandelion seeds in nature, the falling velocity of the as-proposed microflier in air can be facily controlled by tailoring the degree of deformation of the “pappus” under light irradiation with different intensities. Importantly, the judiciously designed microflier is able to achieve a mid-air flight above a light source with a sustained flight time of ~ 8.9 s and a maximum flight

height of ~350 mm thanks to the unique dandelion-like 3D structures. Unexpectedly, the resulting microflier is found to exhibit light-driven upward flight accompanied by autorotating motion, and the rotation mode can be customized in either a clockwise or counterclockwise direction by engineering the shape programmability of bimorph soft actuator films. We sincerely believe that the publication of this exciting work in *Nature Communications* would have a broad impact on a wide range of audiences.

“1) The authors declare that one of the novelties for this research is the direct observation of a separated vortex ring above the microflier by flow visualization experiment. However, in Supplementary Movie 5 and Figure S19, no separated vortex ring appeared (there is just some air fluctuation). There is a unique definition about SVR, Nature 562, 414-418 (2018), by using stagnation point. Without such PIV characterization, one cannot say the SRV exists.”

Our response: Thanks for kind comments. The definition of separated vortex rings (SVR) in the literature (Nature 2018, 562, 414-418) is as follows: *“The attached and separated vortex rings can be distinguished based on the position of the upstream stagnation point (z_{su}): if $z_{su} > 0$, the vortex is separated; otherwise it is attached.”* Per suggestion, we have recorded new PIV videos with high resolution to directly observe SVRs above the microfliers with different fixed “pappus” opening angles, as shown in **Supplementary Movie 5**. Moreover, we have used Matlab software to process the video and obtain the stacked images of air vortex rings above the microfliers (**Supplementary Fig. 28**). The size of the vortex rings is found to increase with an increase in the “pappus” opening angle, and the observation of an SVR is very clear above the microflier with a “pappus” opening angle of 120°.

Per suggestion, **Supplementary Movie 5** and **Supplementary Fig. 28** have been added to the revised supplemental information, and the corresponding descriptions have been added to the revised manuscript as follows:

On page 7 of revised manuscript: *“To further observe the flow distribution around the microfliers and how updraft would be interacting with thick boundary layers surrounding each fiberglass strand, we used same-sized microfliers with different fixed “pappus” opening angles of 0°, 60°, 90°, and 120° to study their interaction with surrounding airflow. Interestingly, a separated vortex ring was observed above the microflier, and the size of the vortex rings is found to increase with an increase in the “pappus” opening angle, and the observation of a separated vortex ring is very clear above the microflier with a “pappus” opening angle of 120°, as shown in Supplementary Fig. 28 and Supplementary Movie 5.”*

Supplementary Fig. 28. PIV characterization showing air vortex rings above the microfliers. *a* Schematic illustration of the air distribution above artificial microflier under light irradiation (Philips infrared lamp, 120 W). Stacked images of air vortex rings above the microfliers with different fixed “pappus” opening angles: *b* 60 °, *c* 90 °, and *d* 120 °. The red rectangle is where the air vortex occurs. *Z* is the distance between artificial microflier and stable vortex rings, here, $Z > 0$ indicates that the vortex is separated.

“2) The flier opens and takes off inside the wind tunnel, this is fine. The heat lamp induces convection and used as a wind tunnel to blow up the structure. This is very confusing. As the very hot lamp functions as a wind, at the same time the light triggers shape morphing. I wonder why the authors should use hot lamp, since there is no way to repeat the same experiments by other researchers—changing another lamp, completely different convection.”

Our response: Many thanks for comments. The lift force of as-proposed microfliers ($F_D = 0.5C_DPAv^2$) is closely related to the projected area of microflier A and updraft velocity v . In our experimental system, it was found that the updraft velocity is highly dependent on the power of light, the position of the light source, and the size of the tunnel that helps to generate a vertical updraft and enhance the light intensity. Therefore, we fixed the size of the tunnel (40 cm in length and 10 cm in diameter) and the position of the light source (located at a distance of 4 cm from the bottom of the tunnel) in the experiment. As a result, light-induced updraft velocity from 0 to 1.25 ms^{-1} can be easily obtained by changing the power of light source (Philips infrared lamp, 0~350 W, wavelength range: 420 nm~1000 nm) as shown in **Supplementary Fig. 27a** of revised supplemental information. It should be noted that light-fueled mid-air flight of microfliers can be also achieved under a simulated solar light (Ceaulight lamp, 300 W, wavelength range: 350 nm~1000 nm), and a sustained flight time of ~2.4 s and a maximum flight height of ~160 mm was observed, as shown in **Supplementary Fig. 26** of revised supplemental information.

As is well known, the aerodynamic flight of microfliers is one of the most complex modes of locomotion, and the conservation of dynamic balance during locomotion is

of particular importance in keeping steady flight, which mainly depends on how to effectively deal with aerodynamic interaction between microflier and airflow. In this work, we believe that light-driven mid-air flight of microfliers benefits from the following two aspects: (1) the as-proposed dandelion-inspired artificial microfliers exhibit symmetrical three-dimensional geometry and symmetrical shape-morphing under appropriate light irradiations; (2) the judiciously designed photoactuation setup is able to generate a vertical updraft and spatially symmetrical light field, and it is worth noting that the light-induced updraft and the light field should be axisymmetric since they are generated from the same light source. When held at the center of the tunnel's outlet, the resulting microflier would symmetrically open its "pappus" and obtain balanced lift force while symmetrically interacting with the vertical updraft. As a result, the microflier is able to achieve a mid-air flight above a light source with a sustained flight time of ~ 8.9 s and a maximum flight height of ~ 350 mm in the experiment. Unexpectedly, the resulting microflier is found to exhibit light-driven upward flight accompanied by autorotating motion, and the rotation mode can be customized in either a clockwise or counterclockwise direction by engineering the shape programmability of bimorph soft actuator films. In theory, the as-proposed microfliers could take off inside a wind tunnel and using a light to control their deformations. In the experiment, it was found that the flight failure greatly increased with a combination of light source and wind tunnel separately, and it should be noted that autorotational flight mode was no longer observed because it is a challenging task to assure the axisymmetry of the vertical wind and light field through aligning the direction of wind flow and the position of the light source.

Per suggestion, **Supplementary Fig. 26** has been added to the revised supplemental information, and the corresponding descriptions have been added in the revised manuscript as follows:

On page 7 of the revised manuscript: *"To achieve light-fueled mid-air flight of the dandelion-inspired artificial microfliers, we judiciously constructed an optical setup composed of an infrared light bulb (Philips infrared lamp, 0~350 W, wavelength range: 420 nm~1000 nm) and a tunnel that helps to generate a vertical updraft and enhance the light intensity in a confined space, as shown in Fig. 4a, Supplementary Fig. 21 and Fig. 22. In the experiment, it was found that the updraft velocity is highly dependent on the power of light, the position of the light source, and the size of the tunnel. Therefore, we fixed the size of the tunnel (40 cm in length and 10 cm in diameter) and the position of the light source (located at a distance of 4 cm from the bottom of the tunnel)."*

"Light-induced updraft velocity from 0 to 1.25 ms^{-1} can be easily obtained through changing the power of light source (Philips infrared lamp, 0~350 W) (Supplementary Fig. 27, Supplementary Movie 4)."

"Importantly, it was found that light-fueled mid-air flight of microfliers can be even achieved under a simulated solar light (Ceaulight lamp, 300 W, wavelength range: 350 nm~1000 nm), and a sustained flight time of ~ 2.4 s and a maximum flight height of ~ 160 mm was observed, as shown in Supplementary Fig. 26."

On page 8 of the revised manuscript: *“As it is well known, the aerodynamic flight of microfliers is one of the most complex modes of locomotion, and the conservation of dynamic balance during locomotion is of particular importance in keeping steady flight, which mainly depends on how to effectively deal with aerodynamic interaction between microflier and airflow. Here, we believe that light-driven mid-air flight of microfliers benefits from the following two aspects: (1) the as-proposed dandelion-inspired artificial microfliers exhibit symmetrical three-dimensional geometry and symmetrical shape-morphing under appropriate light irradiations; (2) the judiciously designed photoactuation setup is able to generate a vertical updraft and spatially symmetrical light field, and it is worth noting that the updraft and the light field should be axisymmetric since they are generated from the same light source. When held at the center of the tunnel’s outlet, the resulting microflier would symmetrically open its “pappus” and obtain balanced lift force while symmetrically interacting with the vertical updraft. As a result, the microflier is able to achieve a mid-air flight above a light source with a sustained flight time of ~ 8.9 s and a maximum flight height of ~350 mm in the experiment. In theory, the as-proposed microfliers would take off inside a wind tunnel while using separately light to control their deformations. In the experiment, however, it was found that the flight failure greatly increased when using a light source and wind tunnel simultaneously, and it should be noted that the autorotation flight mode was no longer observed because it is a challenging task to assure the axisymmetric of the vertical wind and light field through aligning the direction of wind flow and the position of the light source.”*

“3) Whether a real dandelion can fly up the built-up light-driven channel? This controlled experiment will help a lot to analyze the flight mechanism of this aircraft more.”

Our response: Thanks for kind suggestion. Light-driven mid-air flight of real dandelion seeds with open and closed pappus has been investigated under light irradiation (Philips infrared lamp, 300 W) as shown in **Supplementary Movie 7** and **Supplementary Fig. 32** of revised supplemental information. The real dandelion seed with an open pappus was found to exhibit light-driven mid-air flight, while the one with a closed pappus could not fly up at all. Compared with the as-proposed artificial microfliers, the autorotation flight mode was rarely observed in the real dandelion seeds. Per suggestion, **Supplementary Movie 7** and **Supplementary Fig. 32** have been added and corresponding descriptions have been added on Page 8 of revised manuscript: *“Likewise, real dandelion seed with an open pappus was found to exhibit light-driven mid-air flight above the light irradiation (Philips infrared lamp, 300 W), but autorotation flight mode was rarely observed in the real dandelion seeds (Supplementary Movie 7 and Supplementary Fig. 32), confirming that shape-programming of soft actuator plays an important role in autorotation motion genesis.”*

Supplementary Fig. 32. *The light-fueled mid-air flight of a real dandelion seed. Real dandelion seeds with a, b open pappus and c, d closed pappus above a light source (Philips infrared lamp, 300 W).*

In Supplementary Movie 5, the open angle of the pappus is different. Is the power of the lamp different? Why is a tunnel needed? If there is not this tunnel, can the microflier fly above the lamp?"

Our response: Thanks for kind comments. In **Supplementary Movie 5**, we used the microfliers with different fixed “pappus” opening angles of 0° , 60° , 90° , and 120° to confirm the existence of a separated vortex ring above the microflier under light irradiation (Philips infrared lamp, 120 W). As we discussed in your question (2), the tunnel helps to generate a vertical updraft and enhance the light intensity (**Supplementary Fig. 22** and **Supplementary Fig. 24**), which play a critical role in light-driven mid-air flight of as-proposed microfliers. In the experiment, we did not observe the light-driven mid-air flight of as-proposed microfliers in the actuation system without tunnel. Per suggestion, **Supplementary Fig. 22** has been added and corresponding description has been added accordingly.

Supplementary Fig. 22. *Light-induced updraft distribution with tunnel and without tunnel above the light irradiation (Philips infrared lamp, 120 W).*

Why did the lighter flier without pappus in Fig. 3d show a faster falling speed than the flier with pappus in Fig. 3b?"

Our response: Thanks for comments. In our experiment, the weights of all microfliers are adjusted to about 4.0 mg by counterweight. Therefore, the slow falling speed of

microflier with pappus in Fig. 3b might result from the drag-enhanced interaction between air and pappus compared with the microflier without pappus in Fig. 3d.

6) As the microflier flies, it leaves the lamp far away. The change of opening angle of the pappus should be demonstrated. The intensity distribution of the lamp needs to be tested. The exact positional relationship of lamp and tube also needs to be described.

Our response: Thanks for kind comments. It is challenging to track the real-time dynamic change in the opening angle of the pappus of microflier during its flight process. The microflier held at on different heights above the tunnel's outlet. The change in the opening angle of the pappus and intensity distribution of the lamp (Philips infrared lamp, 300 W) have been demonstrated in **Supplementary Fig. 24** of revised supplemental information. Moreover, we also tested the opening angle and light intensity at the same position when there is no tunnel as a controlled experiment. Per suggestion, **Supplementary Fig. 24** has been added to the revised supplemental information, and the exact positional relationship of lamp and tunnel has been addressed accordingly.

Supplementary Fig. 24. Light intensity and “pappus” opening angle of artificial microfliers with tunnel and without tunnel at different heights (Philips infrared lamp, 300 W).

7) In all the Figures, the airflow seems to pass through the middle of the tube of the flier. The airflow should be more acting on the pappus.

Our response: Thanks for kind comments. The diameter of the tubular-shaped bimorph soft actuator in the microflier is 3.5 mm as shown in **Supplementary Fig. 12** of revised supplemental information. As shown in **Supplementary Fig. 29** of revised supplemental information, we observed the air vortex rings above the artificial microflier with a fixed “pappus” opening angle of 120° under (Philips infrared lamp, 120 W). There is no significant difference in the size and shape of air vortex ring for the microfliers with blocked tube and open tube, which indicates that the airflow should

be more acting on the pappus and limited amount of airflow passing through the tube of the microflier. Per suggestion, **Supplementary Fig. 29** has been added in the revised supplemental information, and the corresponding description has been added on Page 7 of revised manuscript: “Moreover, there is no significant difference in the size and shape of air vortex ring for the microfliers with blocked tube and open tube, which indicates that the airflow should be more acting on the pappus and limited amount of airflow passing through the tube of the microflier (Supplementary Fig. 29)”

Supplementary Fig. 29. The air vortex rings above the artificial microflier with a fixed “pappus” opening angle of 120° under light irradiation (Philips infrared lamp, 120 W). (a) Tubular-shaped bimorph soft actuator is blocked. (b) Tubular-shaped bimorph soft actuator is open.

8) No inset chart is found in Fig.2e. Under the same light intensity, why the temperature curves and values in Fig. 2e and Fig. 2d are different?

Our response: Thanks for kind comments. Per suggestion, inset chart has been added in Fig. 2e of revised manuscript. Fig. 2d indicates the maximum temperature variation of bimorph soft actuator film under different light intensities when 808 nm NIR light is turned on for 8 s and off for 16 s, which is used to verify the high photothermal conversion efficiency of ultrasmall AuNRs. Fig. 2e shows the maximum bending angle and corresponding temperature change of the soft actuator as a function of time (NIR, 808 nm, 150 mW cm^{-2}), where the bending angle reaches a maximum value within 2 s.

“9) (I) In Fig. 2b, the description of color bar is missed. (II) In Supplementary Movie 2, the fall process was too fast to be seen clearly. (III) In Fig. 3f, the position of the flier should also be marked in the chart by dots. (IV) The experiment is done with microflier instead of dandelions. It is not proper to use the schematic pictures of dandelions to replace the microflier throughout many Figures in the main text. The light-driven deformation in (V)Fig. 3a should correspond to the expanded pappus on the right, not the left. (VI) In the cycle experiment as shown in Fig. 2g, it seems intuitively that each set of data does not reach 1000 times. It is recommended to display all the cycle data in some separated wide graphs and place it in supplemental information to show more data details.”

Our response: Many thanks for kind suggestions. The following revisions have been made accordingly:

(I) In the caption of Fig. 2b, the following description has been added: “*The color bar corresponds to the color distribution of force magnitudes, where the red areas represent tensile force, whereas blue areas indicate the force with the opposite direction to tensile force.*”

(II) To observe the falling process in **Supplementary Movie 2** more clearly, the playback speed has been changed from 0.5 times to 0.25 times.

(III) In Fig. 3f, the position of the microflier has already been replaced by dots in the charts.

(IV) We have used schematic microfliers to replace the schematic dandelion seeds, as shown in Figures 1e, 1f, 3a, 4b, 4c, 4f of revised manuscript.

(V) The expanded pappus has been rearranged on the right of Fig 3a.

(VI) The complete cycle data of as-proposed microfliers upon different light densities and frequencies have been added in **Supplementary Fig. 11** of revised supplemental information.

Supplementary Fig. 11. Repeated actuation of soft bimorph actuators under different frequencies and light intensities (NIR, 808 nm). In each cycle, the irradiation time accounts for half of the cycle time.

With the changes and point-by-point response to respected reviewers’ comments, we hope that the revised manuscript is now acceptable for publication.

Your kind consideration of this revised manuscript will be much appreciated.

As always, thank you very much for great help and support. Ling

Ling WANG, Ph.D.
Professor, School of Materials Science and Engineering
Tianjin University, Tianjin, 300350, China
[Email: lwang17@tju.edu.cn](mailto:lwang17@tju.edu.cn); Homepage: www.wanglinglab.com

Wei FENG, Ph. D
Professor, School of Materials Science and Engineering,
Tianjin University, P. R. China

REVIEWER COMMENTS

Reviewer #1 (Remarks to the Author):

I appreciate the authors addressing many of my concerns in the previous review and adding a number of new results.

I have a few remaining questions/concerns.

1) What does a 600 degree branding angle mean? Is the y-axis in Supplementary Fig. 10 mislabeled?

2) The authors say “found to quickly open its “pappus” from a closed state to an open state upon being exposed to natural sunlight ($\sim 80 \text{ mW cm}^{-2}$) at room temperature”. Can you please quantify “quickly” for natural sunlight?

3) Supplementary Fig 20 shows that the difference in the terminal velocity in the two states significantly reduces with even a payload of around 24 mg (which is very light for adding any electronics). I think the authors should explicitly add this as a limitation in the discussion section, since it is a very important aspect that should not be left to the supplementary information. I would strongly recommend moving this figure to one of the main figures since this is a very important piece of information that is essential to understanding this system.

4) The current limitation discussion is a bit too positively framed. I would recommend explicitly saying: “our current design has the following limitations. 1) It does not support in-device control of when the device changes states, 2) the terminal velocity difference reduces even with a payload of around 24 mg. and 3)...” Explicitly stating this will help a reader understand more clearly what is left to be done in this domain.

5) Finally, I recently came across this paper that was published in Dec 2022, which does claim to do something very similar. I think it will be help for a reader for the authors to cite and compare with this prior work.

Dandelion-Inspired, Wind-Dispersed Polymer-Assembly Controlled by Light, *Advanced science*, 2022

<https://onlinelibrary.wiley.com/doi/full/10.1002/adv.202206752>

Reviewer #2 (Remarks to the Author):

Thank you for addressing my comments and modifying the main manuscript and supporting information. I appreciate the authors' effort to address my concerns in their response to initial review. In conclusion, I have no further suggestions for improvement.

Reviewer #3 (Remarks to the Author):

The authors have addressed all of my previous comments. I recommend the acceptance of current version.

Our point-by-point response to the respected referees' comments and the changes made to the revised manuscript are as follows.

Reviewer #1:

"I appreciate the authors addressing many of my concerns in the previous review and adding a number of new results. I have a few remaining questions/concerns."

Our response: We greatly appreciate the respected reviewer for highly positive comments on our manuscript. In this work, we demonstrate light-driven dandelion-inspired microfliers based on ultralight and super-sensitive tubular-shaped bimorph soft actuators. Like dandelion seeds in nature, the falling velocity of the as-proposed microflier in air can be facilely controlled by tailoring the degree of deformation of the "pappus" under light irradiation with different intensities. Importantly, the judiciously designed microflier is able to achieve a mid-air flight above a light source with a sustained flight time of ~ 8.9 s and a maximum flight height of ~ 350 mm thanks to the unique dandelion-like 3D structures. Unexpectedly, the resulting microflier is found to exhibit light-driven upward flight accompanied by autorotating motion, and the rotation mode can be customized in either a clockwise or counterclockwise direction by engineering the shape programmability of bimorph soft actuator films. We sincerely believe that the publication of this exciting work in *Nature Communications* would have a broad impact on a wide range of audiences.

"1) What does a 600 degree bending angle mean? Is the y-axis in Supplementary Fig. 10 mislabeled?"

Our response: Many thanks for kind comment. In our manuscript, the bending angle is defined as the out-of-plane shape deformation angle measured from a fixed end to the distal tip segment after actuation. As shown in the inset chart of revised Fig. 2c and Supplementary Fig. 10c, when soft actuator is exposed to NIR light irradiation with low light intensity, the angle change is below 360° , here, angle change is " θ "; when soft actuator is exposed to a higher light intensity (e.g. 150 mW cm^{-2}), the maximum angle change will exceed 360° , so the angle change can be also expressed as $(360^\circ + \theta)$. Per suggestion, Fig. 2c and Supplementary Fig. 10c have been revised in revised manuscript and revised supplemental information.

On page 3 of the revised manuscript:

Fig.2c Experimental data and theoretical analysis of the bending angle as a function of the temperature change. The inset chart depicts the bending angle for the bimorph soft actuator film.

When the bending angle does not exceed 360° , it is “ θ ”; When the bending angle exceeds 360° , it is “ $(\theta+360^\circ)$ ”.

On page 10 of the revised supplemental information.:

Supplementary Fig.10c. The measured maximum bending angle for bimorph soft actuator as a function of NIR light intensity (808 nm, 10–200 mW cm⁻²). When the bending angle does not exceed 360° , it is “ θ ”; When the bending angle exceeds 360° , it is “ $(\theta+360^\circ)$ ”.

“2) The authors say “found to quickly open its “pappus” from a closed state to an open state upon being exposed to natural sunlight ($\sim 80 \text{ mW cm}^{-2}$) at room temperature”. Can you please quantify “quickly” for natural sunlight?”

Our response: Many thanks for kind suggestion. In our experiment, microfliers are found to deform from a closed state to an open state within a time of $\sim 2.5 \text{ s}$ when microfliers are exposed to natural sunlight ($\sim 80 \text{ mW cm}^{-2}$) at room temperature. Per suggestion, the following sentence has been added in the revised manuscript.

On page 4 of the revised manuscript: “The resulting microflier was also found to quickly open its “pappus” from a closed state to an open state within $\sim 2.5 \text{ s}$ upon being exposed to natural sunlight ($\sim 80 \text{ mW cm}^{-2}$) at room temperature.”

“3) Supplementary Fig 20 shows that the difference in the terminal velocity in the two states significantly reduces with even a payload of around 24 mg (which is very light for adding any electronics). I think the authors should explicitly add this as a limitation in the discussion section, since it is a very important aspect that should not be left to the supplementary information. I would strongly recommend moving this figure to one of the main figures since this is a very important piece of information that is essential to understanding this system.”

Our response: Per suggestion, the figure showing the terminal falling velocity of artificial microflier with different payloads under light irradiation has been added as a Fig. 3g in the revised manuscript. The corresponding limitation has been addressed in the discussion section accordingly.

On page 9 of the revised manuscript: “However, the artificial microfliers proposed in this research are still at a very preliminary stage and some critical issues and limitations need to be addressed: (1) The difference in terminal velocity of microfliers between the open state and the closed state is found to reduce even with a payload of around 24 mg. (2).....”

“4) The current limitation discussion is a bit too positively framed. I would recommend explicitly saying: “our current design has the following limitations. 1) It does not support in-device control of when the device changes states, 2) the terminal velocity difference reduces even with a payload of around 24 mg. and 3)...” Explicitly stating this will help a reader understand more clearly what is left to be done in this domain.”

Our response: Per suggestion, the limitation discussions have been reframed in the discussion section of revised manuscript as follows:

On page 9 of the revised manuscript: *“However, the artificial microfliers proposed in this research are still at a very preliminary stage and some critical issues and limitations need to be addressed: (1) The difference in terminal velocity of microfliers between the open state and the closed state is found to reduce even with a payload of around 24 mg. (2) The current design has limited flight control capability because it does not yet support in-device control of the microflier when changing its state. (3) The lack of an end-to-end system integrated with multifunctional sensors (pressure, temperature, and humidity sensors) and an electronic microcontroller results in a microflier that cannot yet demonstrate flight attitude feedback and environment monitoring.”*

“5) Finally, I recently came across this paper that was published in Dec 2022, which does claim to do something every similar. I think it will be help for a reader for the authors to cite and compare with this prior work.

Dandelion-Inspired, Wind-Dispersed Polymer-Assembly Controlled by Light, Advanced science, 2022”

<https://onlinelibrary.wiley.com/doi/full/10.1002/advs.202206752>

Our response: Thanks for comments. The reference (10.1002/advs.202206752) has been cited as Ref. 34 in the previously revised manuscript. In their work, the authors report the design and fabrication of a dandelion-inspired artificial disperser with a two-dimensional structural symmetry based on the rectangular light-responsive liquid crystalline elastomer films attached with fabric filaments, where the lift-off action was observed under the combined assistance of wind flow and light irradiation³⁴. It should be pointed that the resulting disperser exhibits some limitations in terms of dispersal performance as a consequence of its two-dimensional structural asymmetry. It is a challenging task to align the direction of wind flow and the position of the light source, which might result in vortex instability and significantly reduce drag force per unit area. As a result, the artificial disperser showed a flight height of less than 25 mm and a short flight time of <1 s under a significant amount of incident power density of 280 mW cm⁻².

Here, we believe that superior light-driven mid-air flight performance of our microfliers benefits from the following two aspects: (1) the as-proposed dandelion-inspired artificial microfliers exhibit three-dimensional structural symmetry and symmetrical shape-morphing behaviors under appropriated light irradiations; (2) the judiciously designed photoactuation setup is able to generate a vertical updraft and spatially symmetrical light field, and it is worth noting that the updraft and the light

field are axisymmetric since they are generated from the same light source. When held at the center of the tunnel's outlet, the resulting microflier would symmetrically open its "pappus" and obtain balanced lift force while symmetrically interacting with the vertical updraft. As a result, the microflier is able to achieve a mid-air flight above a light source with a sustained flight time of ~8.9 s and a maximum flight height of ~350 mm in the experiment. In theory, our as-proposed microfliers would also take off inside a wind tunnel while using separately light to control their deformations. In the experiment, however, it was found that the flight failure greatly increased when using a light source and wind tunnel simultaneously, and it should be noted that autorotation flight mode was no longer observed because it is a challenging task to assure the axisymmetry of the vertical wind and light field through aligning the direction of wind flow and the position of light source.

Per suggestion, the corresponding descriptions have been added in the revised manuscript as follows:

On page 8 of the revised manuscript: *"Recently, a dandelion-inspired artificial disperser with a two-dimensional structural symmetry was reported based on the rectangular light-responsive liquid crystalline elastomer films attached with fabric filaments, where the lift-off action was observed under the combined assistance of wind flow and light irradiation³⁴. It should be pointed that the resulting disperser exhibits some limitations in terms of dispersal performance as a consequence of its two-dimensional structural asymmetry. It is a challenging task to align the direction of wind flow and the position of the light source, which might result in vortex instability and significantly reduce drag force per unit area. As a result, the artificial disperser showed a flight height of less than 25 mm and a short flight time of <1 s under a significant amount of incident power density of 280 mW cm⁻²."*

Reviewer #2:

"Thank you for addressing my comments and modifying the main manuscript and supporting information. I appreciate the authors' effort to address my concerns in their response to initial review. In conclusion, I have no further suggestions for improvement."

Our response: We are trully appreciated that the respected reviewer has kindly recommended our manuscript to be accepted for publication. We sincerely believe that the publication of this exciting work in *Nature Communications* would have broad impact on a wide range of audience.

Reviewer #3:

"The authors have addressed all of my previous comments. I recommend the acceptance of current version."

Our response: We greatly appreciate the respected reviewer for recommending the acceptance of current version in *Nature Communications*.